# OVERTHINKING THE TRUTH: UNDERSTANDING HOW LANGUAGE MODELS PROCESS FALSE DEMONSTRATIONS

**Danny Halawi**[*]**, Jean-Stanislas Denain**[*]**, and Jacob Steinhardt**
UC Berkeley
`{dhalawi,js_denain,jsteinhardt}@berkeley.edu`

## ABSTRACT

Modern language models can imitate complex patterns through few-shot learning, enabling them to complete challenging tasks without fine-tuning. However, imitation can also lead models to reproduce inaccuracies or harmful content if present in the context. We study harmful imitation through the lens of a model's internal representations, and identify two related phenomena: *overthinking* and *false induction heads*. The first phenomenon, overthinking, appears when we decode predictions from intermediate layers, given correct vs. incorrect few-shot demonstrations. At early layers, both demonstrations induce similar model behavior, but the behavior diverges sharply at some "critical layer", after which the accuracy given incorrect demonstrations progressively decreases. The second phenomenon, false induction heads, are a possible mechanistic cause of overthinking: these are heads in late layers that attend to and copy false information from previous demonstrations, and whose ablation reduces overthinking. Beyond scientific understanding, our results suggest that studying intermediate model computations could be a promising avenue for understanding and guarding against harmful model behaviors.[1]

## 1 INTRODUCTION

A key behavior of modern language models is context-following: large-scale transformer models are able to infer and imitate the patterns in their prompt (Brown et al., 2020). At its best, this allows language models to perform well on benchmarks without the need for fine-tuning (Rae et al., 2021; Hoffmann et al., 2022; Chowdhery et al., 2022; Srivastava et al., 2022). This has led researchers to study how context affects few-shot performance (Min et al., 2022; Kim et al., 2022; Xie et al., 2021; Zhao et al., 2021) as well as the internal mechanisms that produce it (Olsson et al., 2022).

However, context-following can also lead to incorrect, toxic, or unsafe model outputs (Rong, 2021). For example, if an inexperienced programmer prompts Codex with poorly written or vulnerable code, the model is more likely to produce poorly written or vulnerable code completions (Jones & Steinhardt, 2022; Perry et al., 2022). Intuitively, the issue is that context-following learns too much—in addition to inferring the overall intent of the in-context task (what code a user is trying to write), it also learns the pattern of user errors and reproduces it, similar to how gradient-based learning algorithms reproduce label errors in their predictions (Sambasivan et al., 2021).

In this work, we seek to better understand harmful context-following. Since models often perform well zero-shot, we conjecture that when presented with a harmful context, the model *knows* the right answer, but imitates and *says* the wrong answer (Meng et al., 2022a). This lead us to study how incorrect imitations emerge over the course of the model's processing, and to look for the model components that cause them.

To investigate this, we set up a contrast task, where models are provided either correct or incorrect labels for few-shot classification (Figure 1, left). We study the difference between these two settings

---

[*]Equal contribution

[1]All code needed to reproduce our results can be found at `https://github.com/dannyallover/overthinking_the_truth`

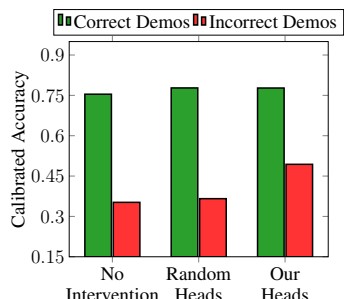

Figure 1: **Left:** Given a prompt of incorrect demonstrations, language models are more likely to output incorrect labels. **Center:** When demonstrations are incorrect, zeroing out the later layers increases the classification accuracy, here on Financial-Phrasebank. **Right:** We identify 5 attention heads and remove them from the model: this reduces the effect of incorrect demonstrations by 32.6% on Financial-Phrasebank, without decreasing the accuracy given correct demonstrations.

by decoding from successively later layers of the residual stream (Nostalgebraist, 2020) (Figure 1, center). Intuitively, this allows us to decode the model's intermediate predictions as it iteratively builds its final output, and to determine which stages of computation propagate the incorrect labels.

We find that correct and incorrect demonstrations yield similar accuracy at early stages of computation, until some "critical layer" at which they sharply diverge. After the critical layer, performance improves given correct demonstrations but drops given incorrect demonstrations. In particular, when demonstrations are incorrect, the neural network "overthinks" (Kaya et al., 2018): stopping the model early increases its accuracy.

We localize overthinking to specific attention heads that attend to and reproduce previous incorrect demonstrations, analogous to the "induction heads" identified in Olsson et al. (2022). These heads are concentrated in the later layers of the model (after the critical layer), perhaps because they attend to complex features (the correctness of an example) that are not present in earlier layers. Removing 5 such heads (1% of heads) reduced the accuracy gap between correct and incorrect prompts by an average of 38.9% over 14 datasets, with negligible effects on the performance given correct prompts (Figure 1, right).

In summary, we found that harmful context-following only appears late in a model's computation, and identified specific attention heads that contribute to these incorrect imitations. More generally, our findings suggest that benign and harmful model behaviors are often processed differently. Indeed, follow-up work (Belrose et al., 2023) has used and extended our insights to detect prompt injection attacks (Perez & Ribeiro, 2022). To proactively understand and reduce harmful model behaviors, researchers should continue to build tools to understand their intermediate computations.

## 2 RELATED WORK

Our work is related to Min et al. (2022), Kim et al. (2022), and Wei et al. (2023), who examine the role of inaccurate demonstrations on model accuracy. Min et al. (2022, figure 4) find that for the pre-trained model GPT-J, the correctness of demonstrations has a large effect on classification accuracy. These works measure the input-output behavior of models on misleading prompts, whereas our work investigates model internals: early-exiting allows us to study how the model builds its representations, and our ablations make it possible to understand the role of specific attention heads.

This high-level perspective matches that of recent work in *mechanistic interpretability* (Cammarata et al., 2021; Geiger et al., 2021; Elhage et al., 2021), which analyzes model internals to reverse engineer the algorithms learned by the network. Mechanistic interpretability techniques have previously been used to study behaviors such as modular arithmetic (Nanda et al., 2023), or factual recall (Meng et al., 2022a;b). However, we take a less "bottom-up" approach than most mechanistic interpretability work: we focus on the role of layers and attention heads, rather than lower-level components such as individual neurons or key, query and value vectors. Moreover, mechanistic interpretability techniques are typically applied to small scale, synthetic tasks, such as indirect object identification (Wang

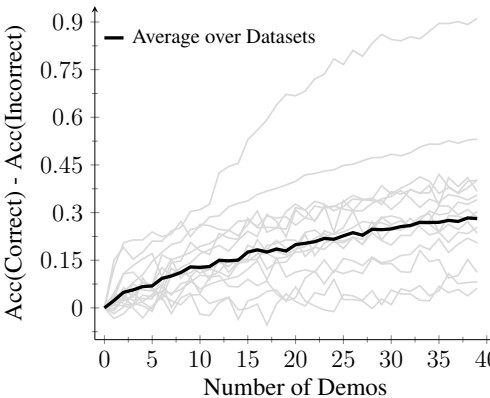 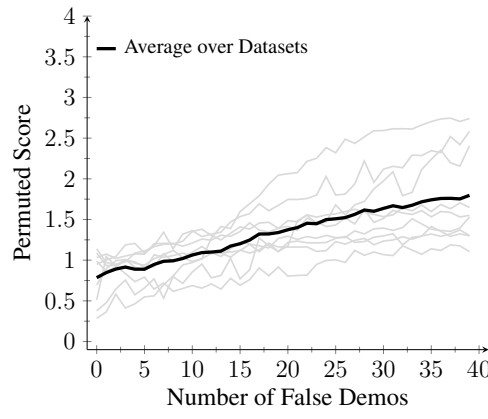

Figure 2: GPT-J behavior in the permuted labels setting (3.1). **Left:** The difference in accuracy between correct and incorrect prompts increases with the number of demonstrations. **Right:** As the number of false demonstrations increases, the model chooses the permuted label $\sigma(\text{class}(x))$ more often than the other labels, rather than making random errors.

et al., 2022). In contrast, we study model behavior across a variety of more realistic tasks, including sentiment analysis, natural language inference, and topic classification.

The literature on early-exiting and overthinking (Kaya et al., 2018; Panda et al., 2015; Teerapittayanon et al., 2017; Figurnov et al., 2017; Hou et al., 2020; Liu et al., 2020; Xin et al., 2020; Zhou et al., 2020; Zhu, 2021; Schuster et al., 2022) also investigates decoding from intermediate layers. These works focus on using early-exiting to improve inference speed, although Mehra et al. (2022) also study the accuracy under distribution shift. In contrast, we use early exiting to scientifically understand the intermediate steps of the model's computation. Moreover, most early exiting methods modify the training process to allow for early exit, or train additional probes to decode intermediate states. In contrast, we use the logit lens (Nostalgebraist, 2020), which does not require any extra training to decode answers from internal representations.

## 3 PRELIMINARIES: FEW-SHOT LEARNING WITH FALSE DEMONSTRATIONS

We begin by introducing the setting we study: few-shot learning for classification, given demonstrations with correct or incorrect labels. Incorrect demonstrations consistently reduce classification performance, which is the phenomenon that we aim to study in this work.

**Few-shot learning.** We consider autoregressive transformer language models, which produce a conditional probability distribution $p(t_{n+1} \mid t_1, ..., t_n)$ over the next token $t_{n+1}$ given previous tokens. We focus on few-shot learning (Brown et al., 2020) for classification tasks: given a task instruction $u$, we sample $k$ demonstrations (input-label pairs) from the task dataset, denoted $(x_1, y_1), ..., (x_k, y_k)$. To query the model on a new input $x$, we use the predictive distribution $p(y \mid u, x_1, y_1, ..., x_k, y_k, x)$.

**Datasets and models.** We consider fourteen text classification datasets: SST-2 (Socher et al., 2013), Poem Sentiment (Sheng & Uthus, 2020), Financial Phrasebank (Malo et al., 2014), Ethos (Mollas et al., 2020), TweetEval-Hate, -Atheism, and -Feminist (Barbieri et al., 2020), Medical Questions Pairs (McCreery et al., 2020), MRPC (Wang et al., 2019), SICK (Marelli et al., 2014), RTE (Wang et al., 2019), AGNews (Zhang et al., 2015), TREC (Voorhees & Tice, 2000), and DBpedia (Zhang et al., 2015). We used the same prompt formats as in Min et al. (2022) and Zhao et al. (2021) (Table 6, 5). For SST-2 we use the 15 prompt formats in Zhao et al. (Table 7). We also considered a toy dataset, Unnatural, that extends a task in Rong (2021). In Unnatural, demonstrations are of the form "[object]: [label]" and the labels are "plant/vegetable", "sport", and "animal".

We evaluated 8 pretrained autoregressive language models: GPT-J-6B (Wang & Komatsuzaki, 2021), GPT2-XL-1.5B (Radford et al., 2019), GPT-NeoX-20B (Black et al., 2022); Pythia models with 410M, 2.8B, 6.9B, and 12B parameters (Biderman et al., 2023); and Llama2-7B (Touvron et al.,

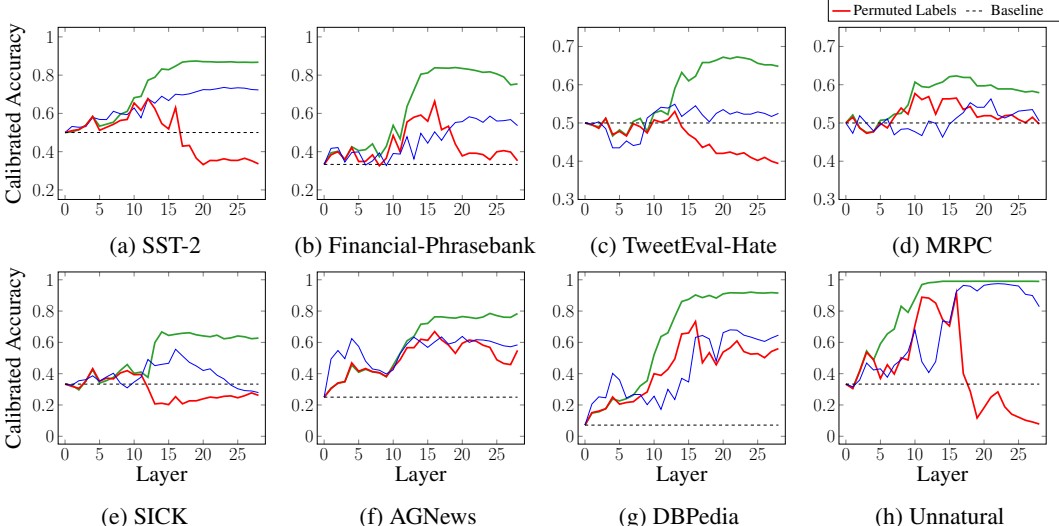

Figure 3: GPT-J early-exit classification accuracies across 6 task categories, given accurate and inaccurate demonstrations (here in the permuted labels setting). Plots are grouped by task type: sentiment analysis (a-b), hate speech detection (c), paraphrase detection (d), natural language inference (e), topic classification (f-g), and a toy task (h). Given incorrect demonstrations, zeroing out all transformer blocks after layer 16 outperforms running the entire model.

2023). We also evaluated instruction-tuned versions of GPT-2-XL (Gallego, 2023), GPT-J-6B (Cloud, 2023) and GPT-NeoX-20B (Clive, 2023).

**Evaluation metrics.** Given our focus on classification tasks, we are interested in how often the model assigns higher probability to the true label than to any other label. However, model predictions can be very unstable with respect to small prompt perturbations (Gao et al., 2021). To mitigate this variability, we measure the *calibrated* classification accuracy (Zhao et al., 2021). Concretely, for a 2-class classification task, we measure how often the correct label has a higher probability than its median probability over the dataset. Assuming the dataset is balanced, which we enforce by sampling demonstration labels with equal probability, this step has been shown to improve performance and reduce variability across prompts. Calibration for multi-class tasks follows a similar procedure, detailed in Appendix A.3.

## 3.1 FALSE DEMONSTRATION LABELS DECREASE ACCURACY

We first set up our contrast task and confirm that the models we study exhibit false context-following behavior. Concretely, we compare the performance of models when the demonstration labels are all correct, i.e. $y_i = \text{class}(x_i)$, and when they are all incorrect, i.e. $y_i = \sigma(\text{class}(x_i))$, for a cyclic permutation $\sigma$ over the set of classes (Figure 1, left). In particular, inputs from the same class are always assigned the same (possibly incorrect) label within each prompt. Because all few-shot labels are chosen according to a permutation of the classes, we call this the permuted labels setting.

For each model and dataset, we sample 1000 sequences each containing $k$ demonstrations and evaluate the model's calibrated accuracy. We sample different demonstrations $(x_i, y_i)$ and label permutations $\sigma$ for every sequence, and vary $k$ from 0 to 40 (from 0 to 20 for GPT2-XL, due to its smaller context size).

Figure 2 (left) shows the difference between GPT-J's calibrated accuracy given correct and incorrect prompts as the number of demonstrations increases. As expected, incorrect demonstrations lead to worse performance, and the accuracy gap tends to increase with $k$ for most datasets. These results are in agreement with Min et al. (2022), who found that incorrect demonstrations decreased GPT-J's performance on classification tasks (Min et al., Figure 4).

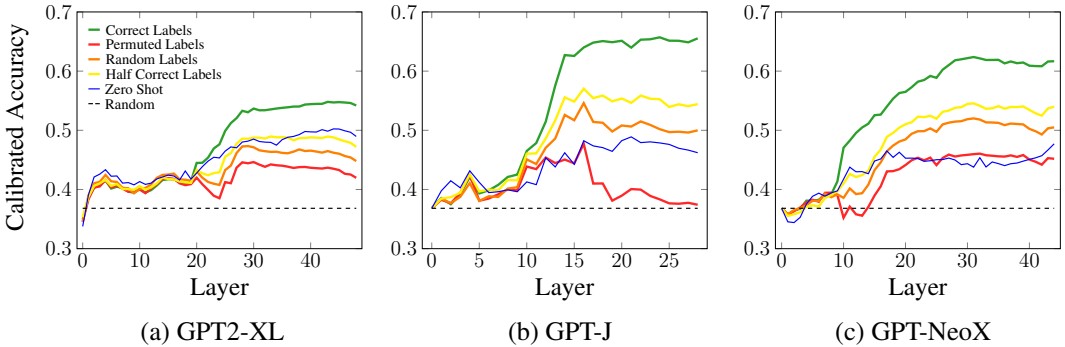

Figure 4: Average calibrated accuracy across 14 tasks for GPT2-XL (a), GPT-J (b), and GPT-NeoX (c). Early-exiting outperforms running the entire model when the demonstrations contain permuted, random, or half correct labels.

Models could lose accuracy by copying the incorrect label, or by becoming confused and choosing random labels. To confirm it is the former, we also measure which labels the model chooses for tasks with more than 2 labels. Specifically, we measure the *permuted score*: how often the model chooses the permuted label $\sigma(\text{class}(x))$ over the other labels. For each dataset, a random classifier would have a permuted score of $\frac{1}{\#\text{labels}}$. To make the results comparable across datasets, we divide the permuted scores by this random baseline. Figure 2 (right) shows these normalized permuted scores for GPT-J on the 9 multi-class datasets in our collection, as well as the average across datasets. The permuted score increases steadily and reaches twice its initial value after 40 demonstrations.

## 3.2 RANDOM AND PARTIALLY CORRECT LABELS LEAD TO LOWER ACCURACY THAN CORRECT LABELS

In the previous subsection, we presented a particular kind of misleading prompt, in which all demonstration labels are chosen according to a permutation of the classes. To study other kinds of misleading prompts, we consider variations on this setup: prompts in which half the demonstrations have correct labels and half have permuted labels (*half correct labels*), and prompts where each demonstration label is chosen at random (*random labels*). These prompts also lead to worse classification accuracy compared to true demonstrations: the accuracy gap for GPT-J at $k = 40$ is $0.15$ for random labels and $0.12$ for half correct labels, which is around half the value for permuted labels ($0.28$).

## 4 ZEROING LATER LAYERS IMPROVES ACCURACY

In this section, to study false context-following, we decode model predictions directly from intermediate layers. This allows us to evaluate the model's performance midway through processing the inputs. On incorrect demonstrations, we find that the model performs *better* midway through processing, especially for GPT-J, and investigate this phenomenon in detail.

**Intermediate layer predictions: the logit lens.** Given an autoregressive transformer language model with $L$ layers, we decode next-token probabilities for each intermediate layer, using the "logit lens" method (Nostalgebraist, 2020). Intuitively, these intermediate distributions represent model predictions after $\ell \in \{1, ..., L\}$ layers of processing.

In more detail, let $h_\ell^{(i)} \in \mathbb{R}^d$ denote the hidden state of token $t_i$ at layer $\ell$, i.e. the sum of everything up to layer $\ell$ in the residual stream. For a sequence of tokens $t_1, ..., t_n \in V$, the logits of the full model's predictive distribution $p(t_{n+1} \mid t_1, ..., t_n)$ are given by

$$[\text{logit}_1, ..., \text{logit}_{|V|}] = W_U \cdot \text{LayerNorm}(h_L^{(n)}),$$

where LayerNorm is the the pre-unembedding layer normalization, and $W_U \in \mathbb{R}^{|V| \times d}$ is the unembedding matrix. The logit lens mimics this operation, but replaces $h_L$ with an intermediate

state $h_\ell$. This yields the intermediate layer distribution $p_\ell(t_{n+1} \mid t_1, ..., t_n)$, defined as

$$[\text{logit}_1^\ell, ..., \text{logit}_{|V|}^\ell] = W_U \cdot \text{LayerNorm}(h_\ell^{(n)}).$$

This provides a measurement of what predictions the model represents at layer $\ell$, without the need to train a new decoding matrix. It can therefore be interpreted as a form of early exiting (Panda et al., 2015; Teerapittayanon et al., 2017; Figurnov et al., 2017).

We compute the intermediate layer distributions $p_\ell$ for our 11 language models, and measure the corresponding calibrated accuracies on the fifteen datasets from Section 3. Figure 4 shows the average accuracy of 3 of our 11 models over the fourteen non-toy datasets as a function of $\ell$, given demonstrations with correct labels, permuted labels, random labels, half correct labels, as well as no demonstrations (we show these results for other models in 7).

**Accurate and incorrect demonstrations sharply diverge at "critical layers".** Given correct demonstrations, the accuracy tends to increase with layer depth. With permuted or random labels, the accuracy follows a similar trend at early layers, but then diverges and decreases at the later layers. This trend is consistent across individual datasets (Figures 3, Figures 9-19).

Moreover, for each model, the accuracies for correct and incorrect prompts diverge at the same layers across almost all datasets: we call these the *critical layers*. For example, for GPT-J, the accuracies diverge between layers 13 and 14 for all but two datasets (Figure 10)[2]. We observe similar results for the other 10 models: for example, for Pythia-6.9B with layer 9, for Llama-2-7B with layers 13 to 17, and for GPT-NeoX-20B-Instruct with layers 10 to 13 (Figures 14, 16, and 19).

**Early-exiting improves classification performance given incorrect demonstrations.** Given incorrect demonstrations, we observe "overthinking": decoding from earlier layers performs *better* than decoding from the final layer. For example, for GPT-J, using $p_{16}$ (the first 16 layers) achieves a better accuracy than the full model on all but one dataset (Figures 3, 4b). Early-exiting also outperforms the full model for our 10 other models: in particular, overthinking does not seem to be affected by model size (Figure 7 (a-d)) or instruction tuning (Figure 7 (e-g)). Furthermore, overthinking is not a result of undertraining. In contrast to our other models, Llama2-7B was trained using the scaling laws from (Hoffmann et al., 2022), yet $p_{19}$ outperforms the full model for all 14 datasets. Finally, early exiting also helps for other misleading prompts: our results were qualitatively similar given random labels and half correct labels (see Figure 4 and 7).

**Ablating attention heads only improves accuracy further.** We hypothesize that correct and incorrect demonstrations diverge at the critical layers because the correctnessness of each demonstration is only encoded after these layers. This would imply that overthinking is caused by the late *attention* layers, which attend back to the late layers of previous demonstrations. To test this, we zero out only the attention heads (and not the MLPs) in late layers. When overthinking is most pronounced (e.g. for GPT-J), we find that ablating just the attention heads has a similar effect to ablating the entire layer, whereas ablating just MLPs has a much smaller effect (Table 2). Since removing only late attention heads recovers almost the full effect of early-exiting, we conclude that these late heads, more than MLPs, are responsible for overthinking. This motivates understanding the attention heads in detail, which we turn to next.

## 5 ZOOMING INTO ATTENTION HEADS

Previously, we found that the gap between true and false demonstrations is predominantly due to attention heads in the later layers of the model. This suggests that false context-following is due to heads attending to complex features in previous demonstrations. In this section, we look for particular heads that are responsible for this context-following behavior.

Drawing from Olsson et al. (2022), we hypothesize that there are *false induction heads* that attend to false labels in similar past demonstrations, and make the model more likely to output them. For example, for the input "beet" in Figure 5, the right-most head attends consistently to the previous incorrect demonstrations of the token "sport".

More formally, we introduce three properties that make a head a false induction head. First, it should be (1) *label-attending*, i.e. concentrate its attention on labels in the previous demonstrations. Second,

---

[2]We formalize this by measuring the layer at which the accuracy gap first reaches half of its final value.

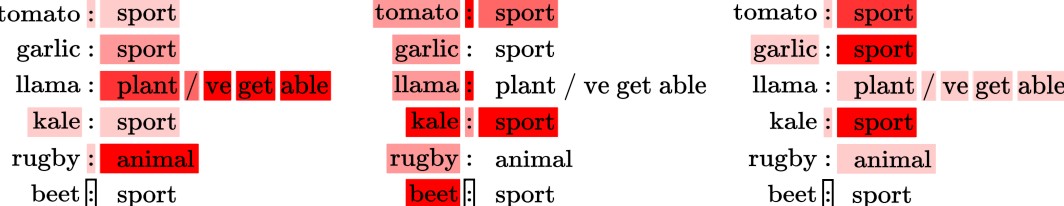

Figure 5: Examples of attention patterns on incorrect demonstrations from the toy Unnatural dataset, for heads that are label-attending but not class-sensitive (Left), heads that are class-sensitive but not label-attending (Center), and heads that are both label-attending and class-sensitive (Right).

it should be (2) *class-sensitive*, meaning it attends specifically to labels that follow inputs from the same class (e.g "tomato", "garlic" and "kale" in Figure 5). Finally, it should be (3) *label-promoting*, meaning it increases the probability of the labels it attends to.

To identify false induction heads, we define a score that quantifies how label-attending and class-sensitive an attention head is (we will return to the label-promoting property at the end of this section). For a sequence of demonstrations $(x_i, y_i)$ and a final input $x$, the **prefix-matching score** ($\text{PM}^h$) of a head $h$ is:

$$\text{PM}^h = \sum_{i=1}^{n} \text{Att}^h(x, y_i) \cdot \mathbf{1}_{\text{class}(x)=\text{class}(x_i)} - \frac{1}{\#\text{labels} - 1} \sum_{i=1}^{n} \text{Att}^h(x, y_i) \cdot \mathbf{1}_{\text{class}(x)\neq\text{class}(x_i)}.$$

This score is high when the head attends strongly to the labels following inputs from class($x$) (first term), and low when the head attends to the labels following other inputs (second term). We compute the prefix-matching score of each head by averaging over incorrect prompts on the Unnatural dataset, and plot the distribution of PM scores across each layer (Figure 6). For all models, the scores remain low at early layers, then increase around the critical layers that we identified in Section 4. This lends correlational support to our hypothesis that false induction heads cause false context-following.

**Ablating false induction heads.** However, we are interested in causal evidence. Therefore, we check whether removing false induction heads reduces false context-following. We select the 5 heads from GPT-J with the highest PM scores, and ablate them by setting their values to zero. We evaluate the resulting lesioned model on all 14 datasets, comparing its layerwise performance to the original model's. As a control baseline, we also perform the same analysis for 5 heads selected at random.

Our ablations significantly increase accuracy given incorrect demonstrations: they reduce the gap between correct and incorrect prompts by an average of 38.9%, with only a small loss in accuracy for correct demonstrations (Table 1). In contrast, ablating random heads barely improves the accuracy given false demonstrations, and sometimes even increases the size of the accuracy gap. These results suggest that false induction heads cause a significant fraction of the false context-following behavior. In addition, since false induction heads were identified using only the toy Unnatural dataset but affect context-following on all datasets, this implies their behavior generalizes across tasks.

**Verifying that our heads are label-promoting.** So far, we have identified label-attending and class-sensitive heads and shown that they contribute to false context-following behavior. To test our initial hypothesis, we next check that they are also label-promoting, i.e. that they increase the probability of the false labels they attend to. We therefore study the outputs of our heads to understand how they affect the residual stream, focusing here on the Unnatural dataset.

We follow the methodology in Wang et al. (2022) to apply the logit lens to each head individually, by applying layer normalization followed by the unembedding matrix to its outputs. This tells us how much the head increases or decreases the intermediate logits of each token. For every head, we define its *false label promoting score* as the difference between the logit increases of the permuted and correct labels. A high score means that the head greatly increase the probability of the permuted label, whereas a score of zero means that it promotes the correct and permuted labels equally.

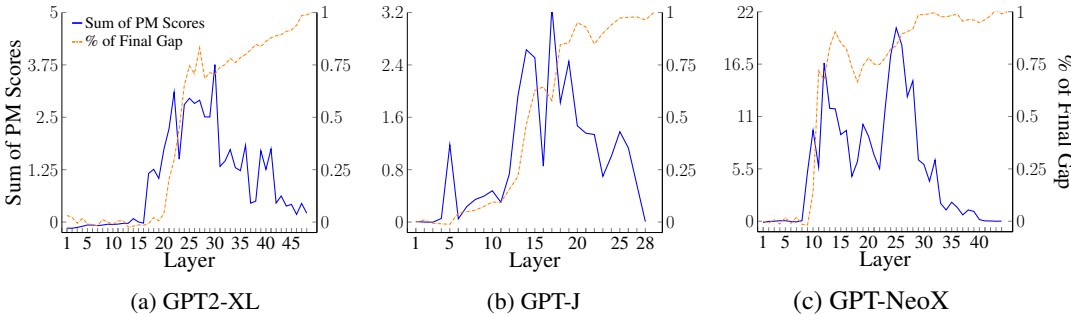

(a) GPT2-XL        (b) GPT-J        (c) GPT-NeoX

Figure 6: Sum of prefix-matching scores for GPT2-XL (a), GPT-J (b), and GPT-NeoX (c) on the toy Unnatural dataset. The prefix-matching scores increase where the accuracy gap (averaged over tasks) between accurate and inaccurate demonstrations emerges.

Our 5 heads have an average false label promoting score of $6.5$: they increase the permuted label logit by $6.5$ more than the correct label on average. In contrast, when sampling 100 sets of 5 random heads, we find an average score of $-0.04$, with a standard deviation of $0.41$. These results confirm that our label-attending and class-sensitive heads are indeed false induction heads.

In summary, our results validate our hypothesis at the beginning of this section: we found a small number of false induction heads in the later layers that contribute to false context-following, by attending to false labels in past demonstrations, and increasing their probability.

## 6 DISCUSSION

In this paper, we studied why language models imitate incorrect demonstrations in their context. By extracting predictions from intermediate model layers, we showed that models *overthink*: given incorrect prompts, the final layers hurt its performance. We then identified a small number of *false induction heads* that attend to and reproduce false information from past demonstrations, and showed via a lesion study that they contribute to incorrect imitation.

**How does the logit lens compare to probing?** Our work, especially Section 4, relies heavily on the "logit lens" (Nostalgebraist, 2020). We find it useful to think of this method in comparison to probing.

If a layer has a high probing accuracy, this means that the correct answer can be decoded from the hidden states. However, this is often a low bar to clear, especially when the classification task is easy and the hidden states are high-dimensional (Hewitt & Liang, 2019). In contrast, if a layer has a high logit lens accuracy, this shows that it encodes correct answers along a direction in the residual stream that the model subsequently decodes from, which is more meaningful. In particular, it implies a high probing accuracy, but the reverse is not necessarily true.

One intermediate between probing and zeroing out later layers is the tuned lens (Belrose et al., 2023): instead of training a new probe for each classification task or directly using the final layer's decoding matrix, Belrose et al. train a single universal "translator matrix" for each layer on a language modelling dataset such as the Pile (Gao et al., 2020). Inspired by our work, Belrose et al. applied the tuned lens to our setup, observing overthinking for additional models such as BLOOM-560M).

**Semantically unrelated labels.**
One hypothesis about the permuted labels setting is that the model simply learns a relabelling of the classes, and is not sensitive to the substance of the incorrect labels. If this were true, we would observe the same logit lens predictions for permuted labels and for semantically unrelated labels (Wei et al., 2023), i.e. labels that have no relation to the task. However, this is not the case: for SST-2, we tried replacing the demonstration labels "Positive" and "Negative" by "A" and "B", and measured the logit lens accuracies in this new setting given incorrect demonstrations (see Figure 10o). While we observe overthinking for related as well as unrelated labels, early-exiting achieves higher than random accuracy for SST-2, but not for its variant. This shows that the ground-truth of demonstration labels is an important factor in our results.

**Realism of our setting.** While we find consistent results across 14 datasets, our experiments are restricted to a specific setting: text classification with a large number of incorrect few-shot examples. Nevertheless, we believe that the permuted labels setting captures important properties of

Table 1: Ablating false induction heads recovers a significant fraction of the accuracy gap between correct and incorrect prompts, without hurting performance given correct demonstrations. We show the percent reduction in the accuracy gap ("Gap") and absolute change in correct prompt performance ("TP") when ablating the 5 false induction heads chosen using the Unnatural dataset ("top") or 5 random heads ("random"). Subscript numbers denote 1 standard error. We bold gap reductions when they are greater for our heads than for the random heads. We show results for one dataset in each task category; full results are in Table 3.

| Dataset | Heads | Permuted Labels | | Half Permuted Labels | | Random Labels | |
|---|---|---|---|---|---|---|---|
| | | $\Delta$ TP ($\uparrow$) | $\Delta$ Gap ($\uparrow$) | $\Delta$ TP ($\uparrow$) | $\Delta$ Gap ($\uparrow$) | $\Delta$ TP ($\uparrow$) | $\Delta$ Gap ($\uparrow$) |
| Poem-Sentiment | top | $1.67_{0.01}$ | $\mathbf{30.76_{0.20}}$ | $2.43_{0.02}$ | $\mathbf{66.36_{0.11}}$ | $1.63_{0.02}$ | $\mathbf{38.97_{0.15}}$ |
| | random | $1.47_{0.01}$ | $4.68_{0.05}$ | $1.27_{0.01}$ | $17.40_{0.07}$ | $0.37_{0.01}$ | $-17.08_{0.12}$ |
| Ethos | top | $-6.00_{0.07}$ | $\mathbf{28.61_{0.06}}$ | $-4.20_{0.06}$ | $-5.21_{0.04}$ | $-3.20_{0.04}$ | $-1.19_{0.01}$ |
| | random | $-3.00_{0.04}$ | $5.97_{0.08}$ | $0.60_{0.01}$ | $7.29_{0.04}$ | $1.40_{0.02}$ | $-2.38_{0.01}$ |
| MRPC | top | $-5.70_{0.02}$ | $\mathbf{89.02_{0.19}}$ | $-1.20_{0.01}$ | $\mathbf{7.69_{0.01}}$ | $0.01_{0.01}$ | $\mathbf{115.79_{0.02}}$ |
| | random | $-3.50_{0.01}$ | $23.17_{0.05}$ | $-1.00_{0.01}$ | $-38.46_{0.05}$ | $0.60_{0.01}$ | $47.37_{0.03}$ |
| SICK | top | $-3.63_{0.03}$ | $\mathbf{15.29_{0.17}}$ | $-9.43_{0.07}$ | $-19.68_{0.14}$ | $-6.20_{0.04}$ | $\mathbf{10.97_{0.08}}$ |
| | random | $2.27_{0.02}$ | $-2.82_{0.04}$ | $-1.80_{0.02}$ | $-10.99_{0.08}$ | $0.13_{0.01}$ | $-0.51_{0.01}$ |
| AGNews | top | $2.40_{0.06}$ | $\mathbf{32.34_{0.20}}$ | $-0.80_{0.02}$ | $\mathbf{46.59_{0.12}}$ | $-1.30_{0.03}$ | $\mathbf{33.77_{0.17}}$ |
| | random | $2.70_{0.06}$ | $-11.06_{0.11}$ | $-1.10_{0.03}$ | $9.09_{0.04}$ | $-1.50_{0.04}$ | $6.49_{0.04}$ |
| Average | top | $-1.26_{0.02}$ | $\mathbf{38.98_{0.16}}$ | $-2.36_{0.01}$ | $\mathbf{15.14_{0.03}}$ | $-1.58_{0.01}$ | $\mathbf{31.47_{0.07}}$ |
| | random | $0.79_{0.02}$ | $3.97_{0.03}$ | $-0.40_{0.01}$ | $-16.50_{0.01}$ | $0.37_{0.01}$ | $-18.74_{0.01}$ |

realistic failure modes. Indeed, humans often err in consistent, systematic ways. For example, an inexperienced coder might consistently use the wrong method name, thereby permuting the method names in their prompts to a code completion model.

Moreover, our findings provide valuable information to understand misleading prompts beyond the permuted labels setting. Indeed, Belrose et al. (2023) drew inspiration from our work to detect another failure of large models: "prompt injection" (Branch et al., 2022). We ran a preliminary analysis of the intermediate predictions in this setting, and found that injected prompts, like incorrect demonstrations, exhibit overthinking (see Figure 26).

**Ablations on true prefix.** Surprisingly, we find that even with correct demonstrations, models have a tendency to overthink. When removing late layers and late attention in GPT2-XL, we observed a net benefit in performance. Furthermore, early exiting at the critical layer improves performance on a majority of datasets across all models. This signifies a potential misalignment between the pretraining objective and the downstream few-shot task, which is an interesting direction for future study.

**Limitations and future work.** Our head ablations do not fully remove the accuracy gap between correct and incorrect demonstrations. This could be because we did not identify some of the model components that cause false context-following. However, there is another possibility: if an attention head's outputs are on average far from zero, zeroing out that head takes the intermediate states off-distribution, which can decrease overall performance. Thus, one promising future direction would be to replace head outputs by their average value, as in Nanda et al. (2023).

Our work relates to mechanistic interpretability, which seeks to reverse engineering model behaviors from a bottom-up understanding of low-level components. In contrast, we embrace a more top-down strategy, extracting predictions from entire layers. This shift not only enhances efficiency, compute, and time, but also allows us to scrutinize model behavior on more realistic tasks. Our results suggest that aberrant and normal model behaviors are often processed differently, so more comprehensively measuring model internals could help us to understand and fix a broad variety of unwanted behaviors.

ACKNOWLEDGEMENTS

Thanks to Erik Jones, Collin Burns, Nora Belrose, Lisa Dunlap, Alex Pan and our anonymous reviewers for helpful comments and feedback. DH was supported by an award from the C3.ai Digital Transformation Institute. JSD is supported by the NSF Division of Mathematical Sciences Grant No. 2031899. JS was supported by the National Science Foundation SaTC CORE Award No. 1804794 and the Simons Foundation.

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

# A APPENDIX

## A.1 LOGIT LENS RESULTS FOR OTHER MODELS

Figure 7, we have plotted the average Logit Lens results for our other models across tasks.

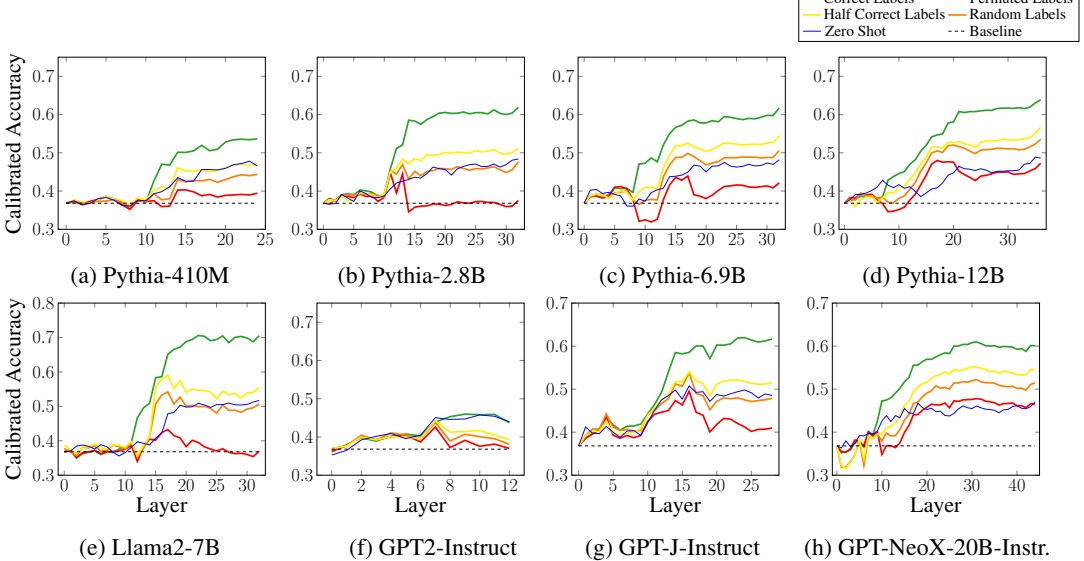

Figure 7: Average calibrated accuracy across 14 tasks for 4 Pythia models of different sizes (a-d), Llama2-7B (e), and instruction-tuned versions of GPT2-XL, GPT-J, and GPT-NeoX-20B (f-h). Early-exiting outperforms running the entire model when the demonstrations contain permuted, random, or half correct labels.

Table 2: Average calibrated accuracy on correct and incorrect labels when running the full model, zeroing out late layers, zeroing out late attention heads (but not MLPs), and zeroing out late MLPs (but not attention heads). We ablate after layer 16 for GPT-J, 30 for GPT2-XL, and 32 for GPT-NeoX. The best and second best ablated accuracy are bolded and underlined respectively. Subscript numbers denote 1 standard error. We find that ablating late attention heads and ablating late layers have similar performance: this suggests that late attention heads play an especially important role in overthinking.

| Model | Permuted Labels | | | | Correct Labels | | | |
|---|---|---|---|---|---|---|---|---|
| | Full Model | Late Heads | Late MLP | Late Layers | Full Model | Late Heads | Late MLP | Late Layers |
| GPT2-XL | $41.97_{1.49}$ | $\mathbf{46.09_{1.49}}$ | $42.88_{1.48}$ | $\underline{44.63_{1.50}}$ | $54.19_{1.51}$ | $\mathbf{54.09_{1.49}}$ | $52.47_{1.51}$ | $\underline{53.68_{1.50}}$ |
| GPT-J | $37.42_{1.47}$ | $\underline{47.58_{1.47}}$ | $37.97_{1.49}$ | $\mathbf{47.72_{1.46}}$ | $65.54_{1.42}$ | $\underline{64.46_{1.39}}$ | $\mathbf{65.84_{1.40}}$ | $64.00_{1.41}$ |
| GPT-NeoX | $45.19_{1.47}$ | $44.44_{1.47}$ | $\underline{44.78_{1.48}}$ | $\mathbf{46.06_{1.49}}$ | $61.68_{1.41}$ | $\underline{60.86_{1.43}}$ | $56.78_{1.34}$ | $\mathbf{62.15_{1.42}}$ |

## A.2 ABLATING ONLY ATTENTION HEADS, OR ONLY MLPS

## A.3 CALIBRATION

For $k$-way tasks, we measure how often the correct label has a higher probabilitiy than the $\frac{k-1}{k}$-quantile of its probability over the dataset. In figure 20, we show the logit lens accuracies of GPT-J over the 16 datasets: although the uncalibrated accuracies at earlier layers are much noisier and occasionally indistinguishable from the baseline accuracy, we also find overthinking on a majority of datasets.

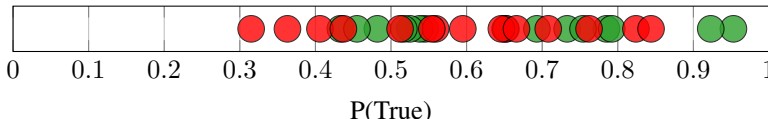

Figure 8: The probability of the label "True" for 30 random test inputs in MRPC. Inputs from the "True" class are marked with green dots, and inputs from the "False" class are marked with red dots. As observed in Zhao et al. (2021), the model can be biased towards one of the labels: here the model tends to assign a higher probability to the "True" label than to the "False" label, for inputs from both classes.

## A.4 LOGIT LENS RESULTS FOR OTHER MODELS ACROSS TASKS

We plot the Logit Lens results across all tasks for all models: GPT2-XL (Figure 9), GPT-J (Figure 10), GPT-NeoX-20B (Figure 11), Pythia-410M (Figure 12), Pythia-2.9B (Figure 13), Pythia-6.9B (Figure 14), Pythia-12B (Figure 15), Llama2-7B (Figure 16), GPT2-Instruct (Figure 17), GPT-J-Instruct (Figure 18), and GPT-NeoX-20B-Instruct (Figure 19).

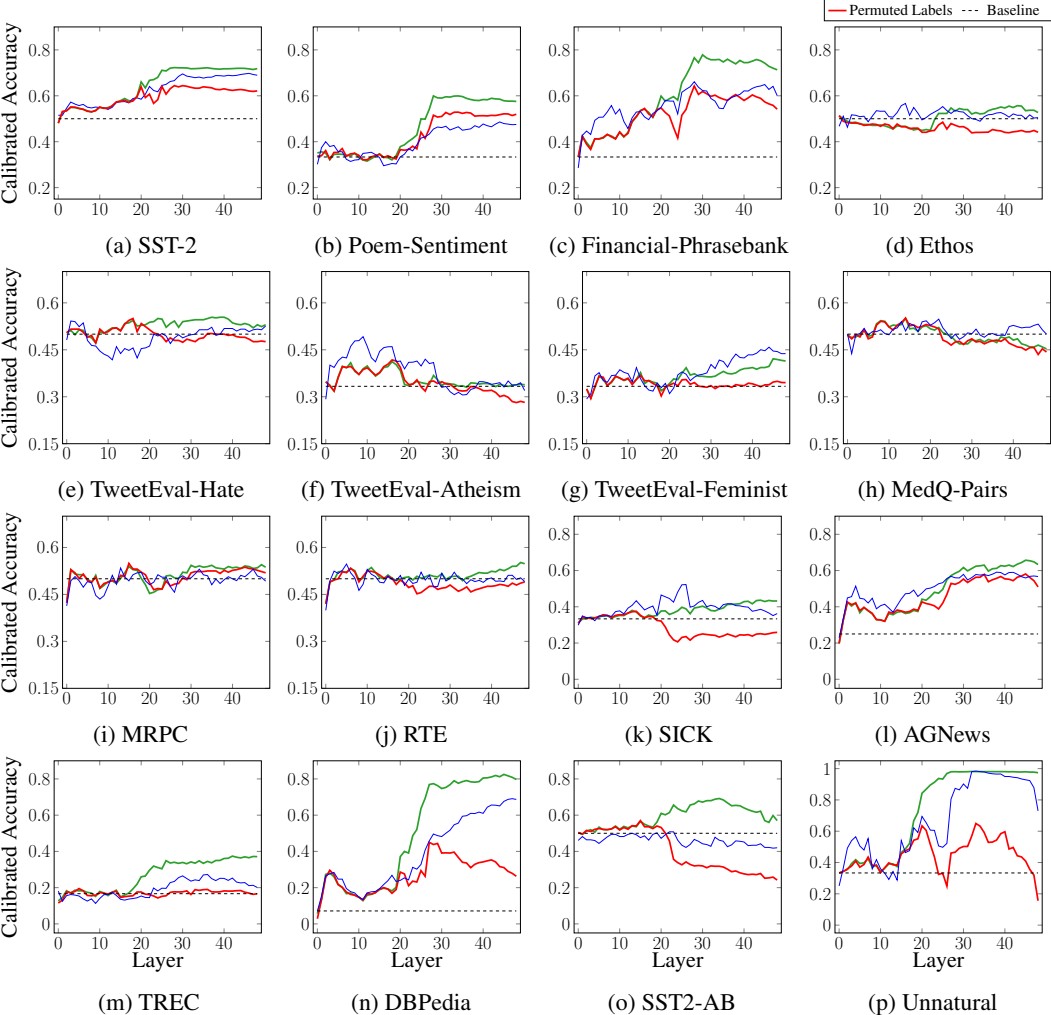

Figure 9: GPT2-XL early-exit classification accuracies across 16 tasks, given correct and incorrect demonstrations. Plots are grouped by task type: sentiment analysis (a-c), hate speech detection (d-g), paraphrase detection (h-i), natural language inference (j-k), topic classification (l-n), and toy tasks (o-p). Given incorrect demonstrations, zeroing out all transformer blocks after layer 36 outperforms running the entire model on 13 out of 16 datasets.

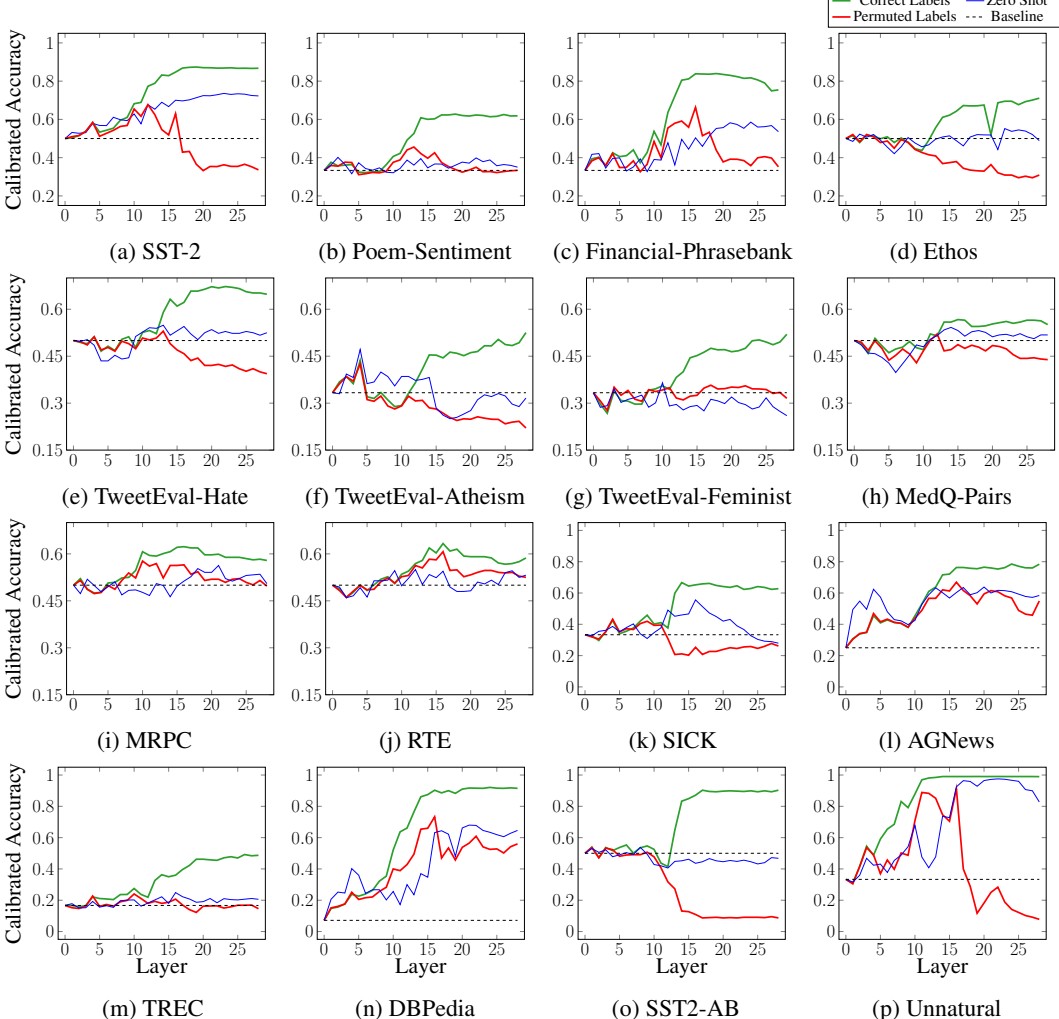

Figure 10: GPT-J early-exit classification accuracies across 16 tasks, given correct and incorrect demonstrations. Plots are grouped by task type: sentiment analysis (a-c), hate speech detection (d-g), paraphrase detection (h-i), natural language inference (j-k), topic classification (l-n), and toy tasks (o-p). Given incorrect demonstrations, zeroing out all transformer blocks after layer 16 outperforms running the entire model on 15 out of 16 datasets.

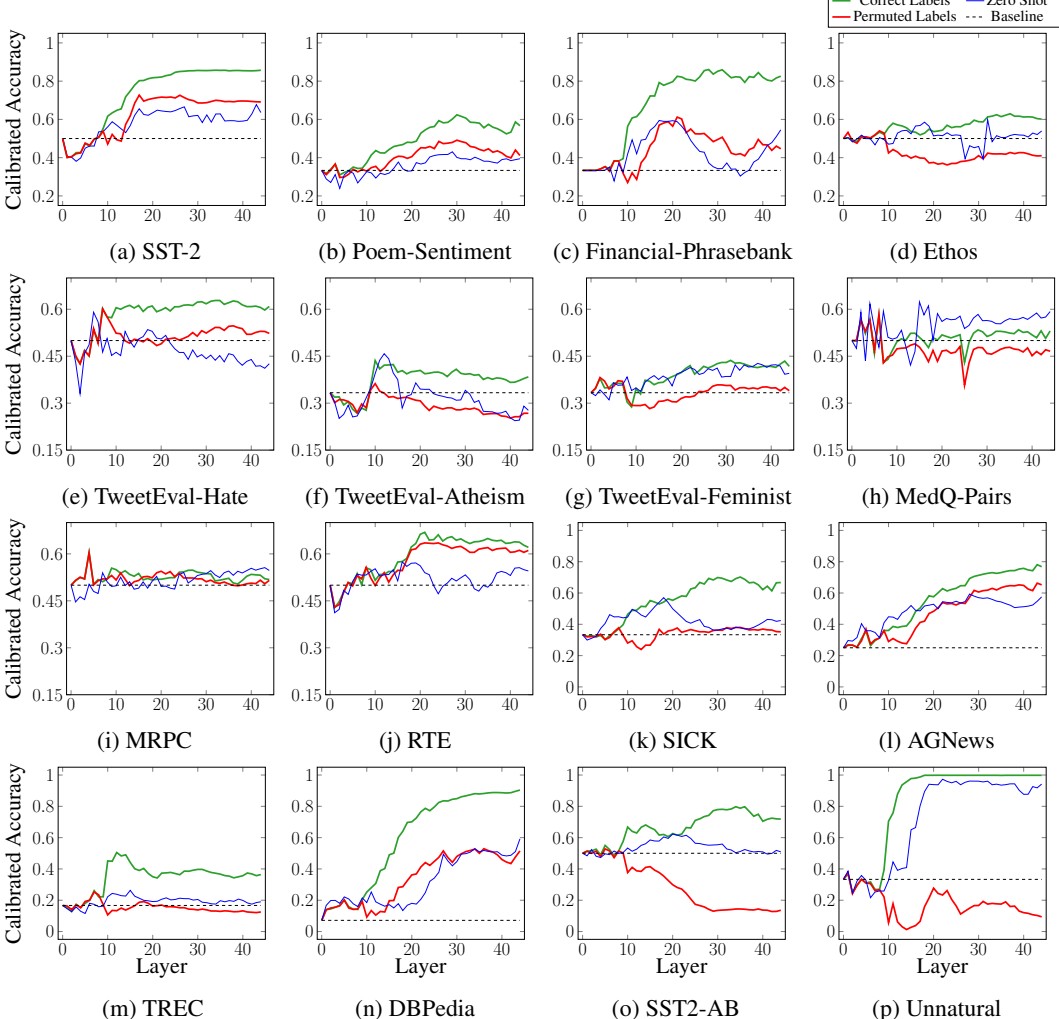

Figure 11: GPT-NeoX-20B early-exit classification accuracies across 16 tasks, given correct and incorrect demonstrations. Plots are grouped by task type: sentiment analysis (a-c), hate speech detection (d-g), paraphrase detection (h-i), natural language inference (j-k), topic classification (l-n), and toy tasks (o-p). Given incorrect demonstrations, zeroing out all transformer blocks after layer 27 outperforms running the entire model on 14 out of 16 datasets.

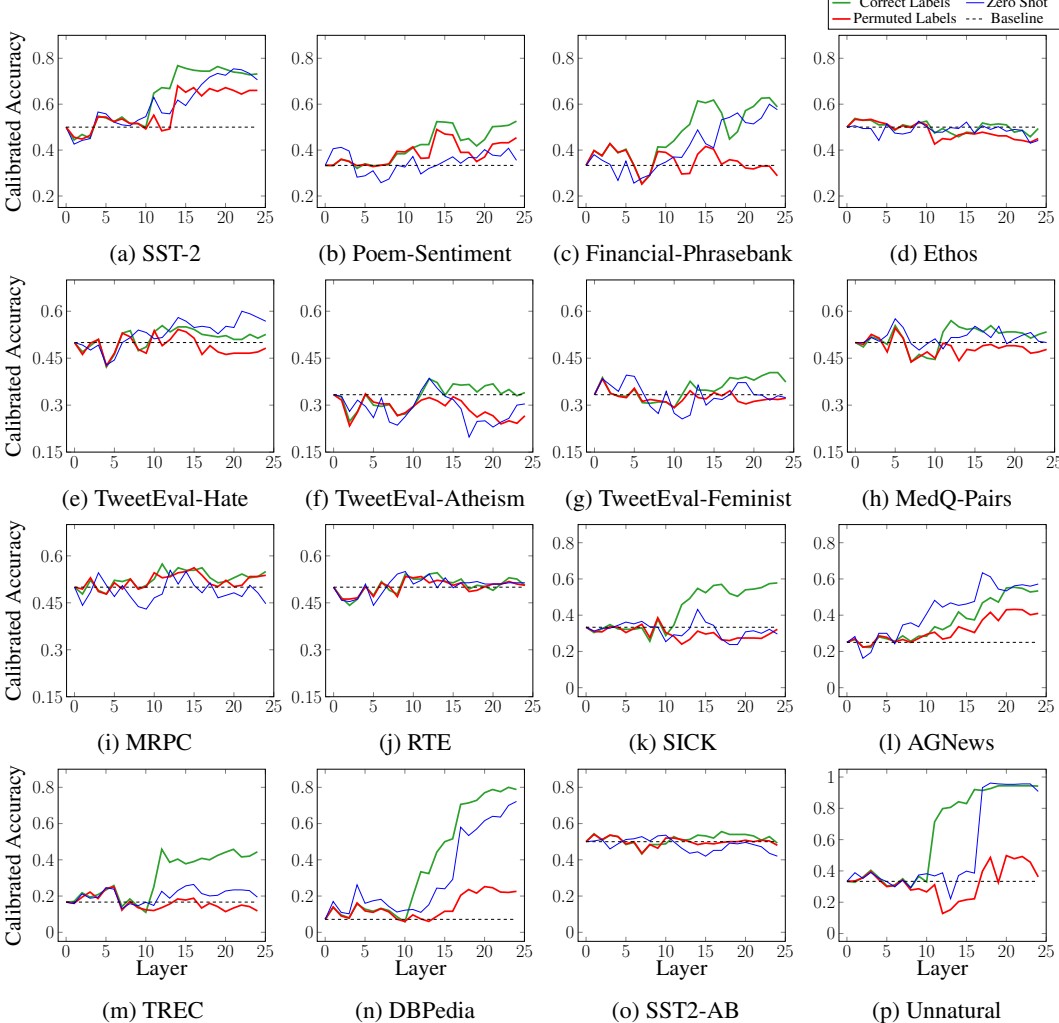

Figure 12: Pythia-410M early-exit classification accuracies across 16 tasks, given correct and incorrect demonstrations. Plots are grouped by task type: sentiment analysis (a-c), hate speech detection (d-g), paraphrase detection (h-i), natural language inference (j-k), topic classification (l-n), and toy tasks (o-p). Given incorrect demonstrations, zeroing out all transformer blocks after layer 14 outperforms running the entire model on 11 out of 16 datasets.

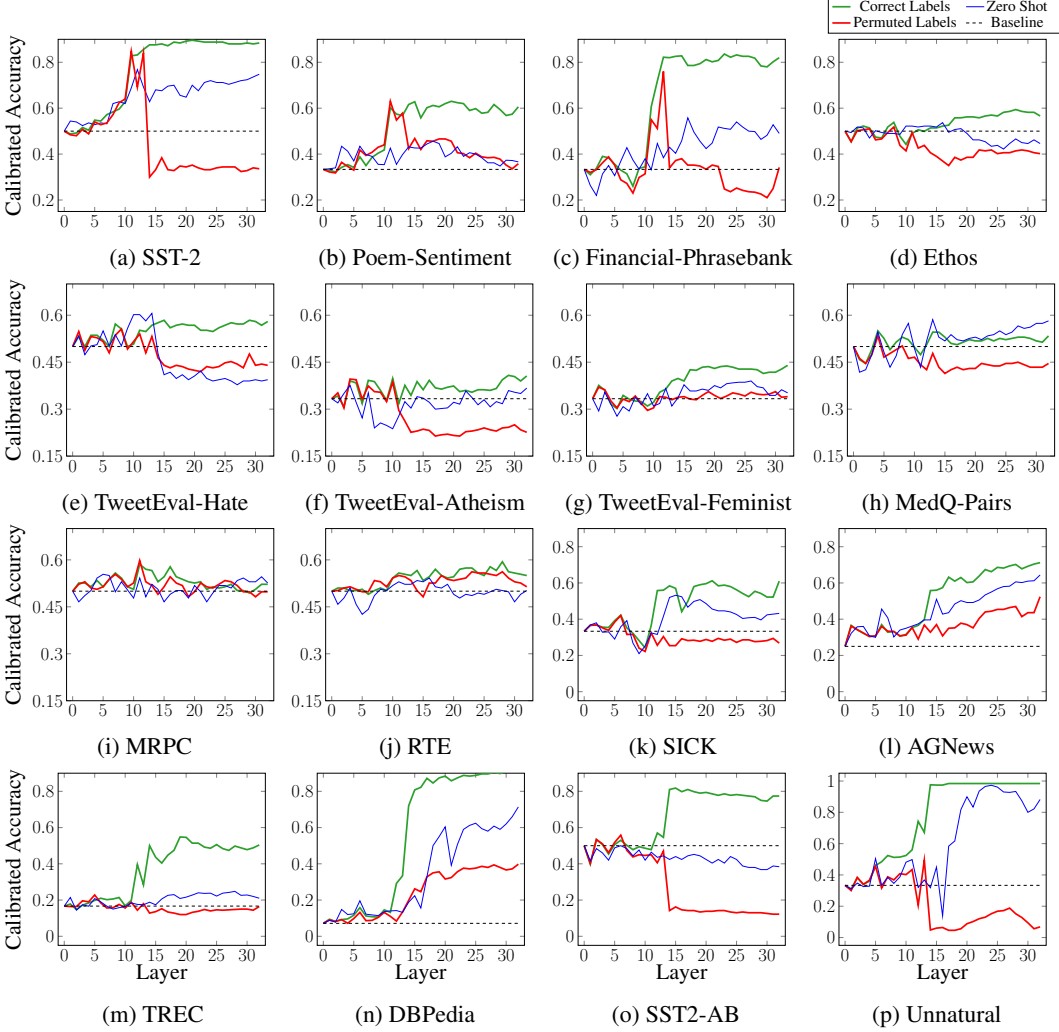

Figure 13: Pythia-2p8B early-exit classification accuracies across 16 tasks, given correct and incorrect demonstrations. Plots are grouped by task type: sentiment analysis (a-c), hate speech detection (d-g), paraphrase detection (h-i), natural language inference (j-k), topic classification (l-n), and toy tasks (o-p). Given incorrect demonstrations, zeroing out all transformer blocks after layer 13 outperforms running the entire model on 12 out of 16 datasets.

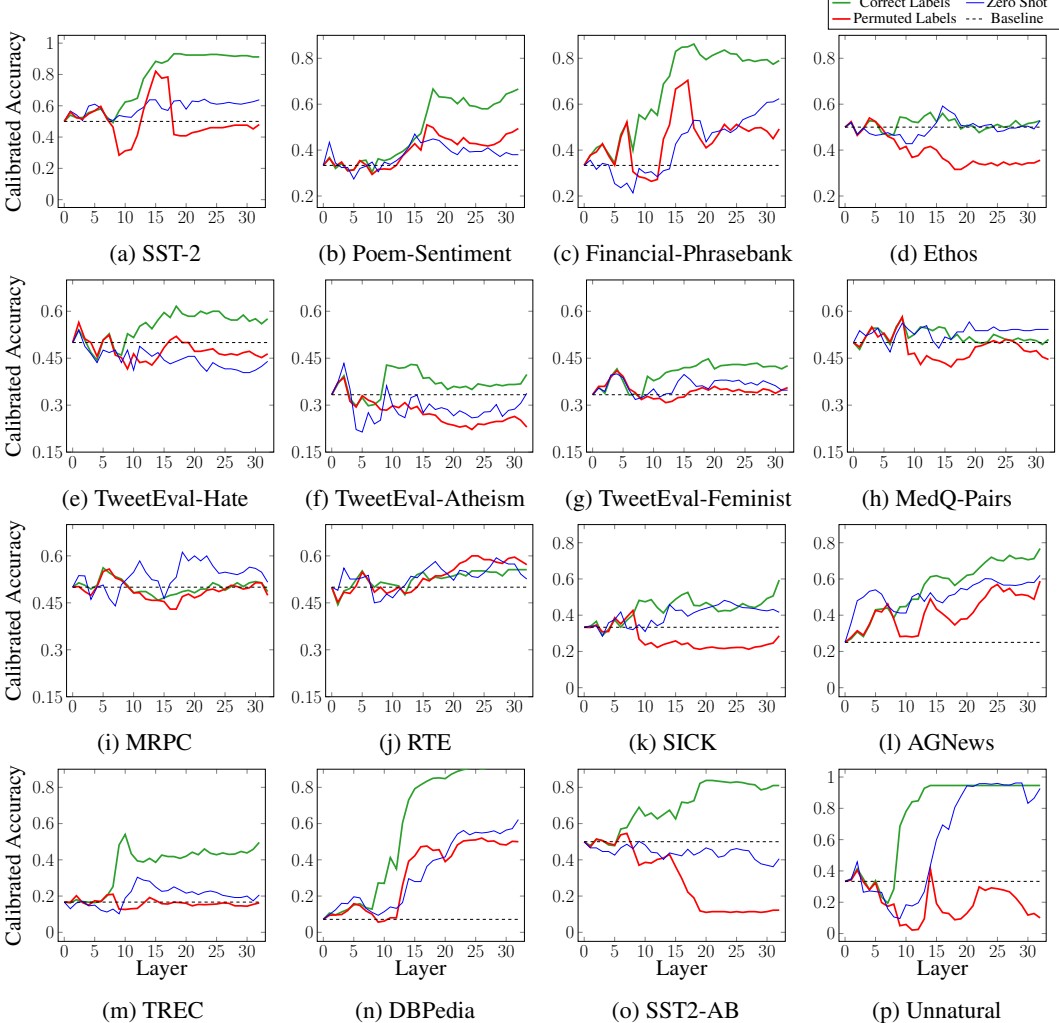

Figure 14: Pythia-6p9B early-exit classification accuracies across 16 tasks, given correct and incorrect demonstrations. Plots are grouped by task type: sentiment analysis (a-c), hate speech detection (d-g), paraphrase detection (h-i), natural language inference (j-k), topic classification (l-n), and toy tasks (o-p). Given incorrect demonstrations, zeroing out all transformer blocks after layer 5 outperforms running the entire model on 11 out of 16 datasets.

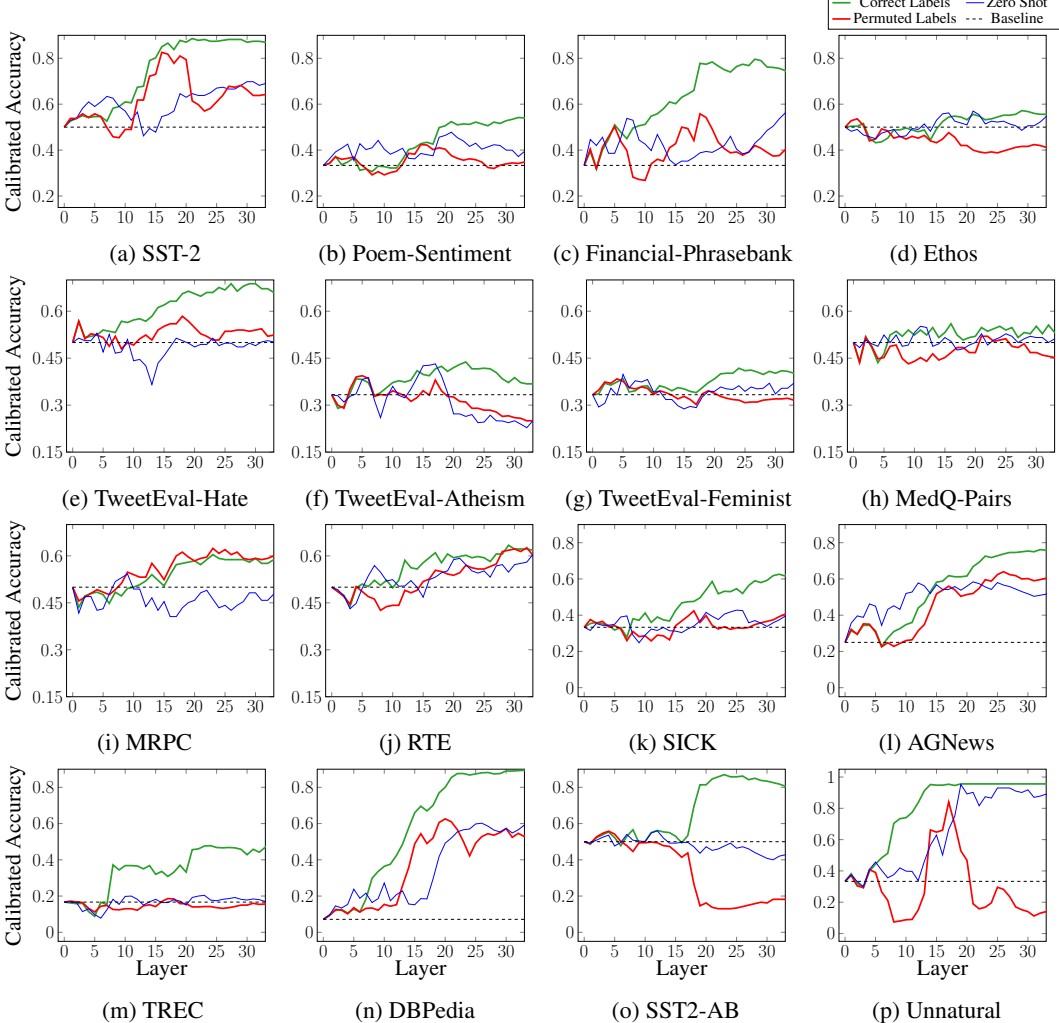

Figure 15: Pythia-12B early-exit classification accuracies across 16 tasks, given correct and incorrect demonstrations. Plots are grouped by task type: sentiment analysis (a-c), hate speech detection (d-g), paraphrase detection (h-i), natural language inference (j-k), topic classification (l-n), and toy tasks (o-p). Given incorrect demonstrations, zeroing out all transformer blocks after layer 16 outperforms running the entire model on 11 out of 16 datasets.

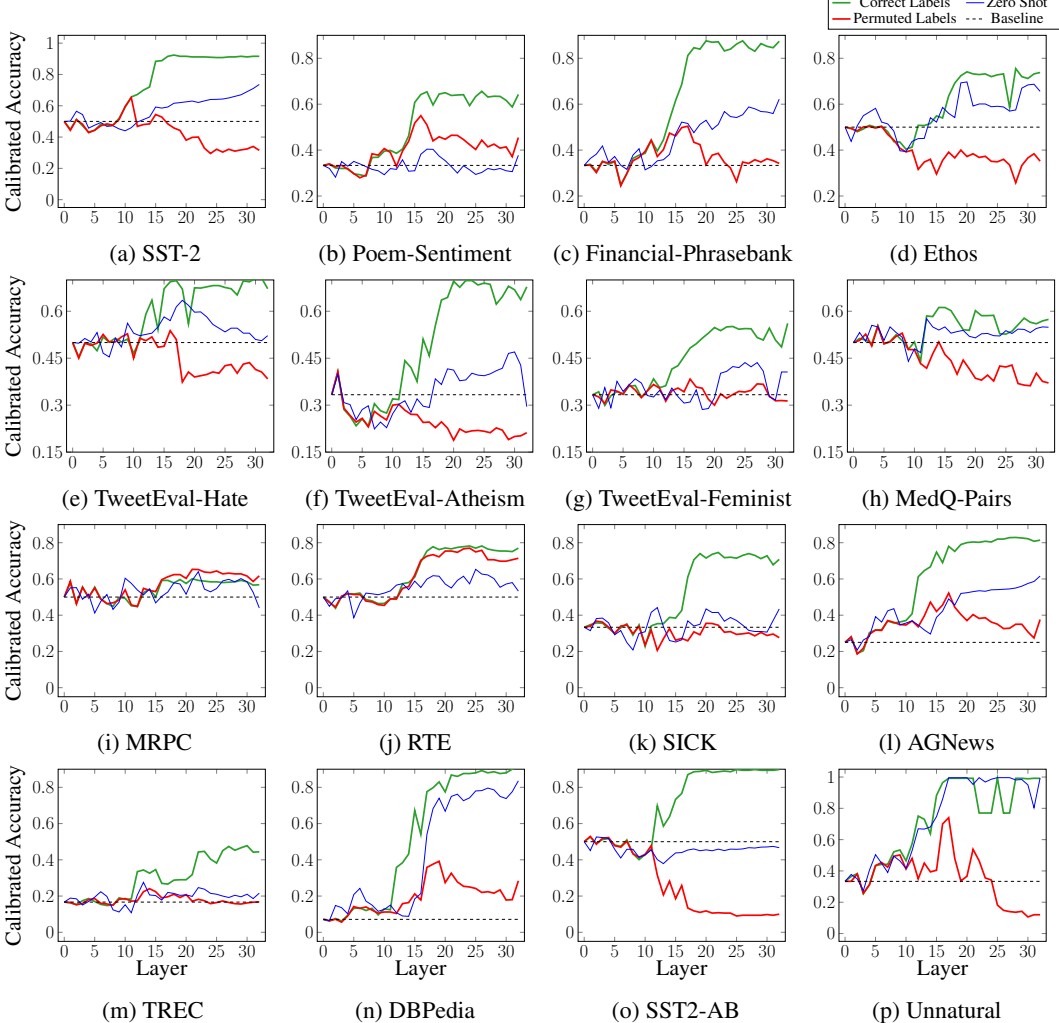

Figure 16: Llama2-7B early-exit classification accuracies across 16 tasks, given correct and incorrect demonstrations. Plots are grouped by task type: sentiment analysis (a-c), hate speech detection (d-g), paraphrase detection (h-i), natural language inference (j-k), topic classification (l-n), and toy tasks (o-p). Given incorrect demonstrations, zeroing out all transformer blocks after layer 19 outperforms running the entire model on 16 out of 16 datasets.

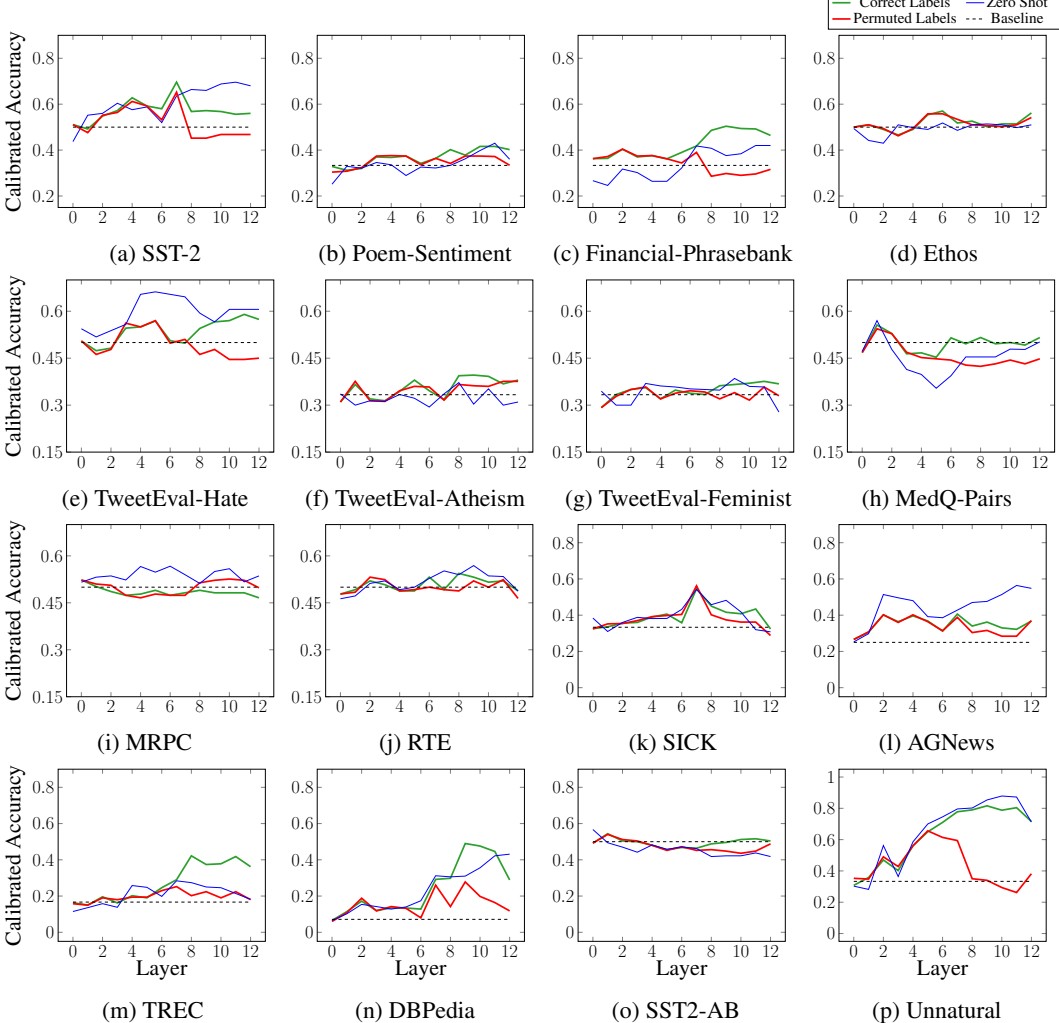

Figure 17: GPT2-Instruct early-exit classification accuracies across 16 tasks, given correct and incorrect demonstrations. Plots are grouped by task type: sentiment analysis (a-c), hate speech detection (d-g), paraphrase detection (h-i), natural language inference (j-k), topic classification (l-n), and toy tasks (o-p). Given incorrect demonstrations, zeroing out all transformer blocks after layer 7 outperforms running the entire model on 12 out of 16 datasets.

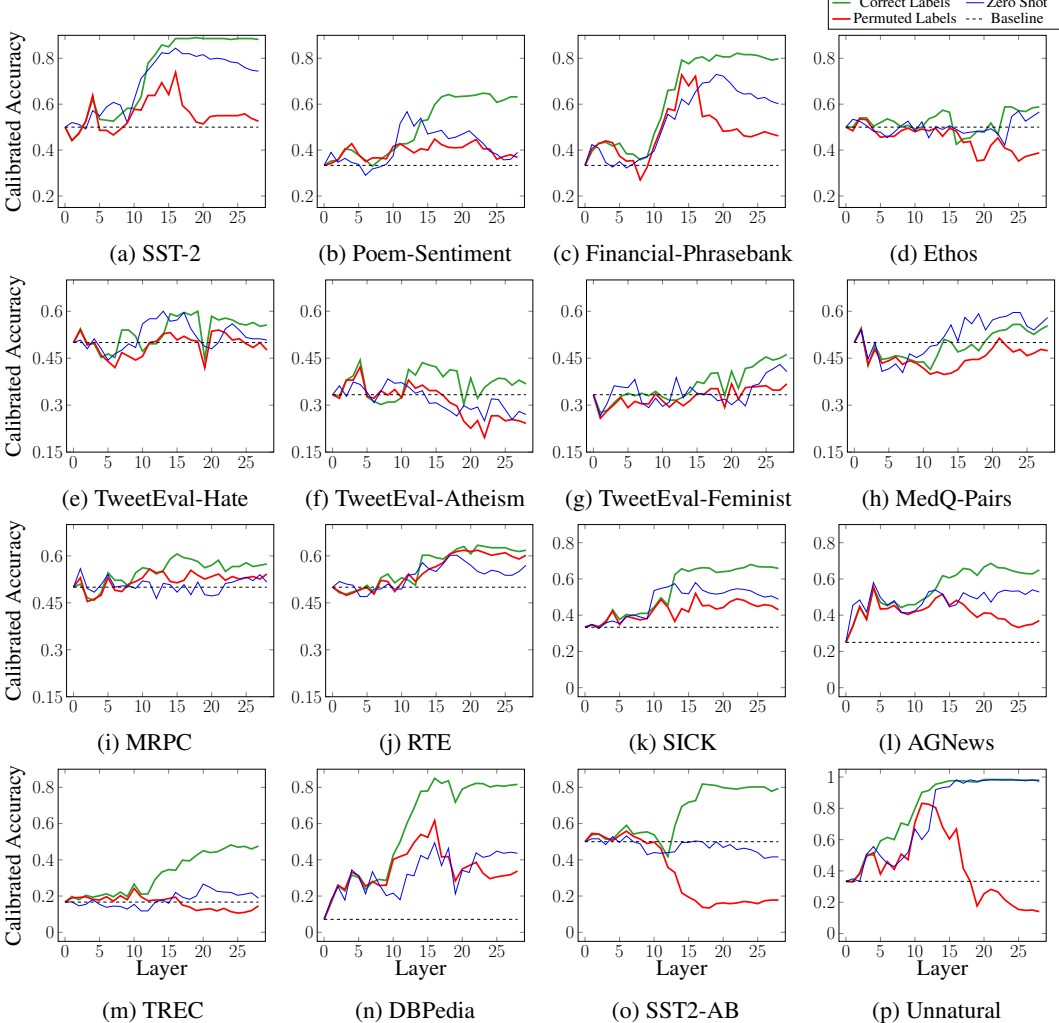

Figure 18: GPT-J-Instruct early-exit classification accuracies across 16 tasks, given correct and incorrect demonstrations. Plots are grouped by task type: sentiment analysis (a-c), hate speech detection (d-g), paraphrase detection (h-i), natural language inference (j-k), topic classification (l-n), and toy tasks (o-p). Given incorrect demonstrations, zeroing out all transformer blocks after layer 17 outperforms running the entire model on 13 out of 16 datasets.

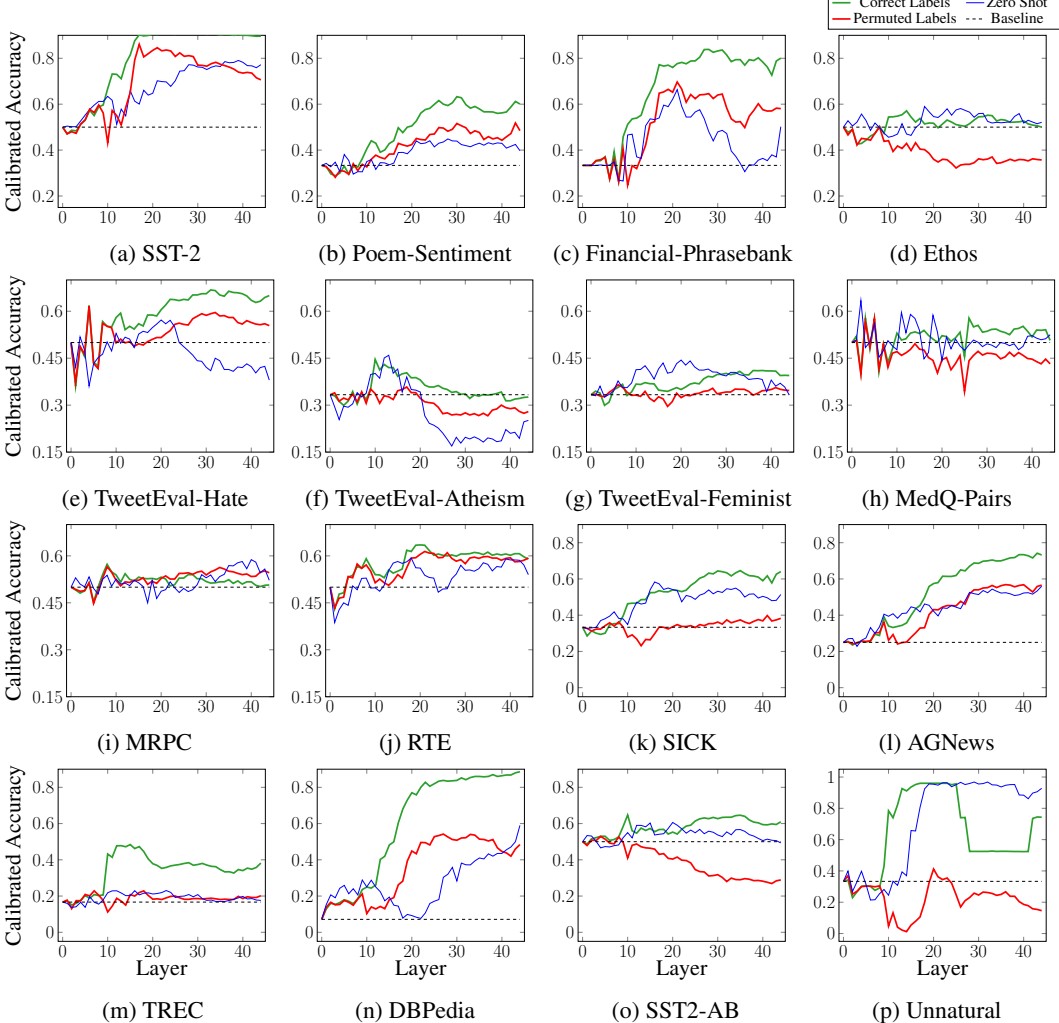

Figure 19: GPT-NeoX-20B-Instruct early-exit classification accuracies across 16 tasks, given correct and incorrect demonstrations. Plots are grouped by task type: sentiment analysis (a-c), hate speech detection (d-g), paraphrase detection (h-i), natural language inference (j-k), topic classification (l-n), and toy tasks (o-p). Given incorrect demonstrations, zeroing out all transformer blocks after layer 32 outperforms running the entire model on 11 out of 16 datasets.

## A.5 Logit lens results for GPT-J without calibration

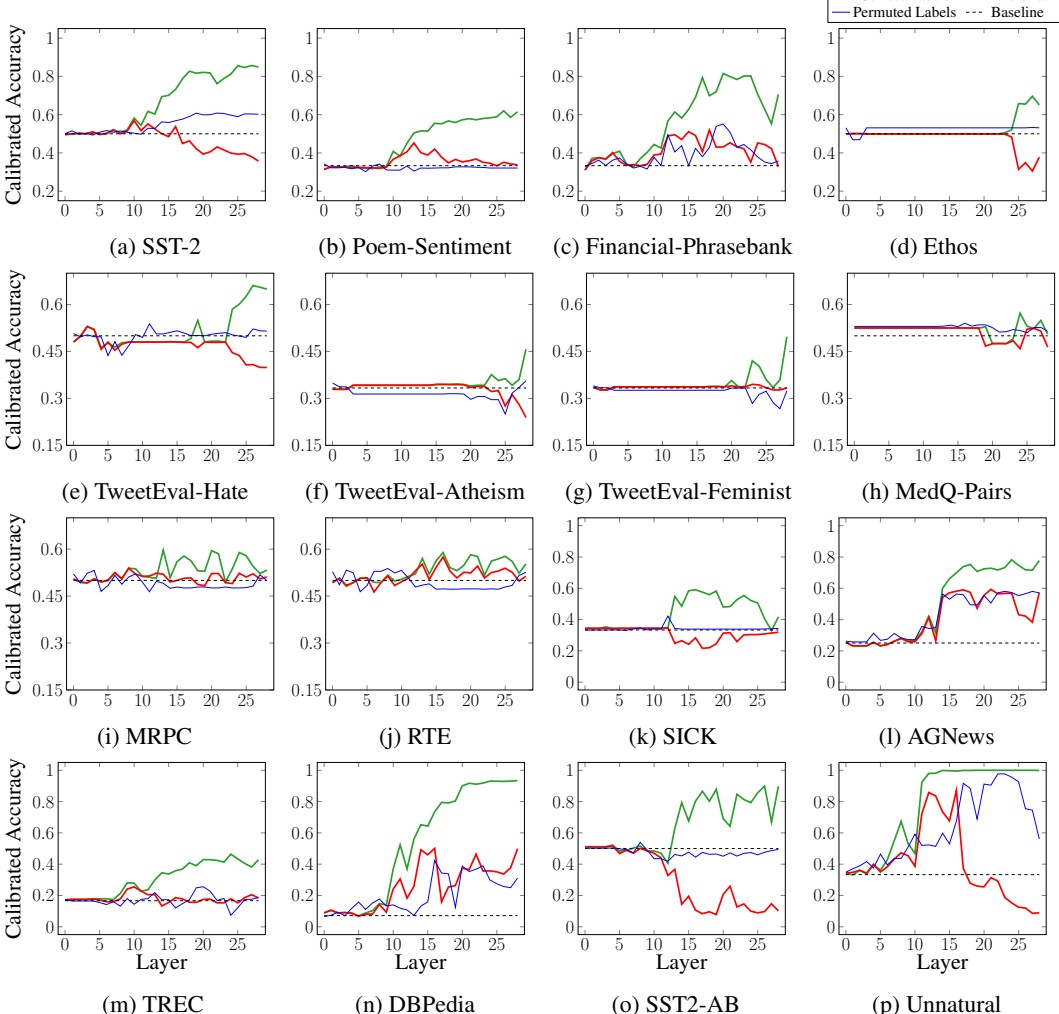

Figure 20: GPT-J early-exit *uncalibrated* classification accuracies across 16 tasks, given correct and incorrect demonstrations. The lack of calibration brings early layer performance to baseline for some datasets, but early-exiting still frequently outperforms running the full model.

## A.6 LOGIT LENS RESULTS FOR EACH SST-2 PROMPT FORMAT

Figure 21: Calibrated Accuracy for all 15 prompt formats for SST-2 (from Zhao et al. (2021)). Given incorrect demonstrations, prompt formats 1, 2, 3, 4, 5, 7, 8, 9, 10, and 13 experience an increase in performance before experiencing a decline. Prompt formats 6, 12, 14, and 15, on the other hand, do not exhibit this effect. Prompt format 11 produces poor performance, given both correct and incorrect demonstrations. See Table 7 for prompt format details.

## A.7 LOGIT LENS RESULTS FOR OTHER METRICS

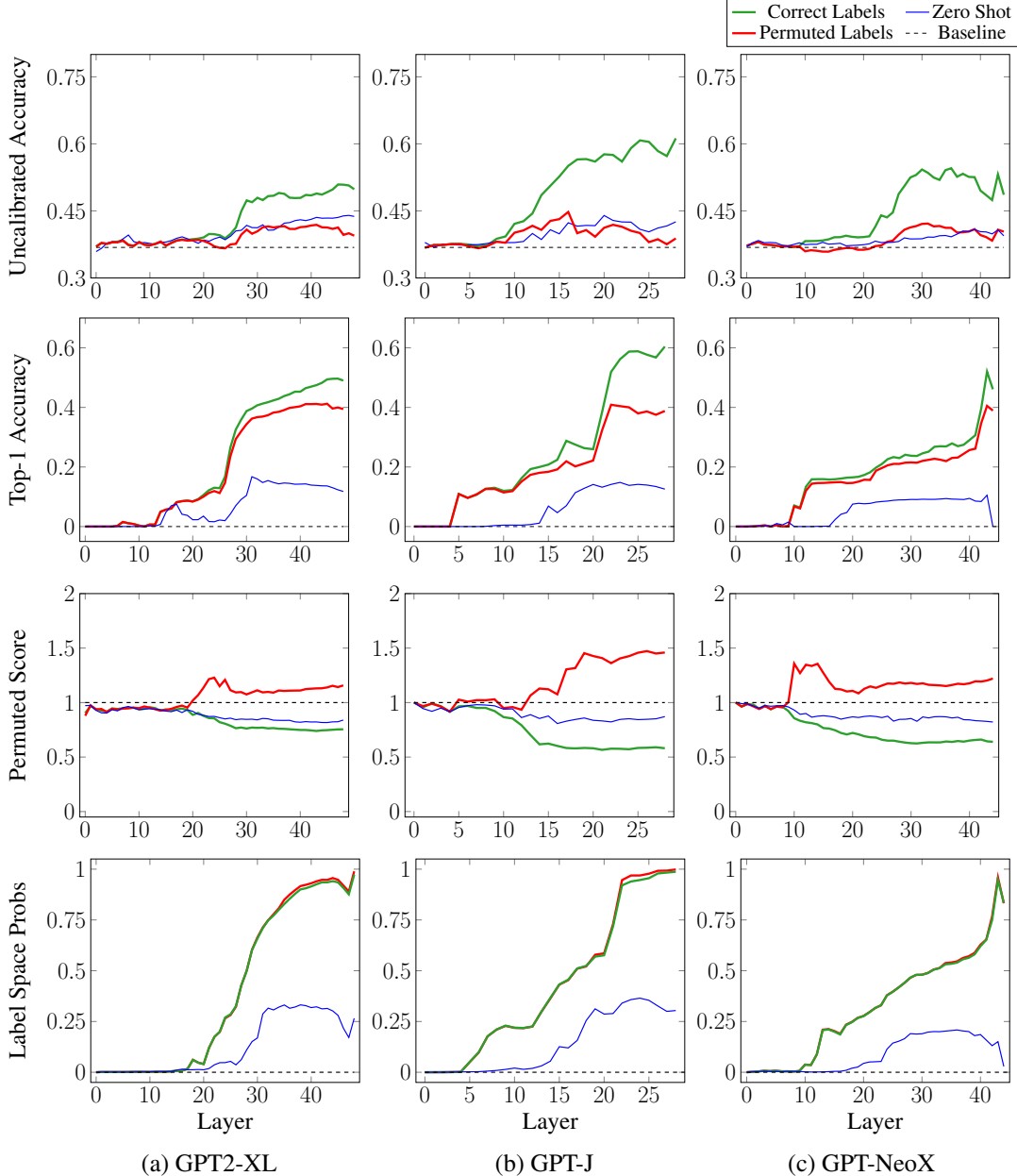

(a) GPT2-XL      (b) GPT-J      (c) GPT-NeoX

Figure 22: Uncalibrated Accuracy (row 1), Top-1 Accuracy (row 2), Permuted Score (row 3), and Label Space Probabilities (row 4) averaged over 14 tasks (9 multi-class tasks for the permuted score). As the label space is learned, we observe the emergence and ensuing increase in the gap in the other metrics.

### A.8 ACCURACY GAP AS A FUNCTION OF $k$ FOR OTHER MODELS

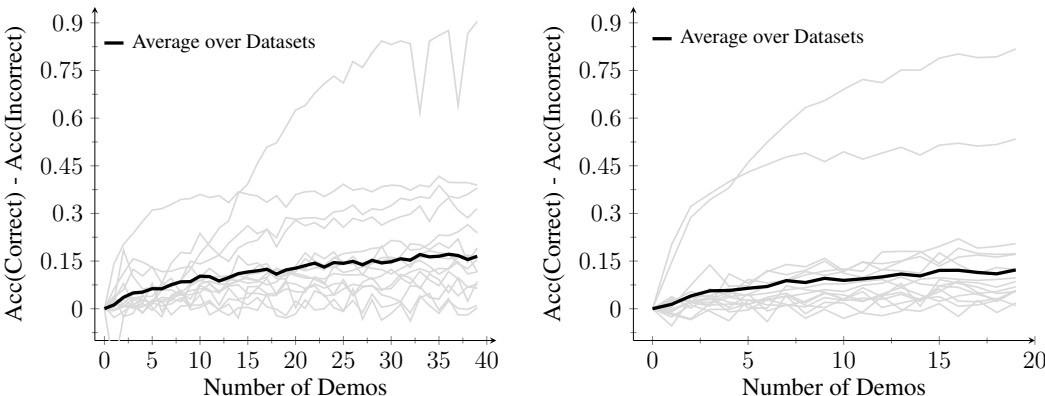

Figure 23: GPT-NeoX (left) and GPT2-XL (right) behavior in the Permuted Labels setting (3.1). The difference in accuracy between accurate and inaccurate prompts increases with the number of demonstrations.

### A.9 FALSE INDUCTION HEAD ABLATION RESULTS ON ALL TASKS

Table 3: Ablating false prefix-matching heads recovers a large fraction of the accuracy gap between true and false prefixes, without hurting performance given true prefixes. We show the percentage reduction of the accuracy gap and percentage change in true prefix performance when ablating the 5 false prefix-matching heads chosen using the Unnatural dataset ("top") or 5 random heads ("random"). We bold gap reductions when they are greater for our heads than for the random heads. Subscript numbers denote 1 standard error.

| Dataset | Heads | Permuted Labels | | Half Permuted Labels | | Random Labels | |
|---|---|---|---|---|---|---|---|
| | | $\Delta$ TP ($\uparrow$) | $\Delta$ Gap ($\uparrow$) | $\Delta$ TP ($\uparrow$) | $\Delta$ Gap ($\uparrow$) | $\Delta$ TP ($\uparrow$) | $\Delta$ Gap ($\uparrow$) |
| *Sentiment Analysis* | | | | | | | |
| SST-2 | top | $5.86_{0.18}$ | $\mathbf{54.56_{0.42}}$ | $3.61_{0.08}$ | $\mathbf{88.71_{0.48}}$ | $4.71_{0.13}$ | $\mathbf{100.62_{0.33}}$ |
| | random | $5.46_{0.16}$ | $-7.94_{0.04}$ | $2.01_{0.02}$ | $23.94_{0.61}$ | $3.71_{0.08}$ | $21.43_{0.59}$ |
| Poem-Sentiment | top | $1.67_{0.01}$ | $\mathbf{30.76_{0.20}}$ | $2.43_{0.02}$ | $\mathbf{66.36_{0.11}}$ | $1.63_{0.02}$ | $\mathbf{38.97_{0.15}}$ |
| | random | $1.47_{0.01}$ | $4.68_{0.05}$ | $1.27_{0.01}$ | $17.40_{0.07}$ | $0.37_{0.01}$ | $-17.08_{0.12}$ |
| Financial-Phrasebank | top | $2.30_{0.05}$ | $\mathbf{32.67_{0.30}}$ | $-2.60_{0.05}$ | $\mathbf{14.72_{0.10}}$ | $1.43_{0.03}$ | $\mathbf{25.34_{0.16}}$ |
| | random | $2.33_{0.05}$ | $5.89_{0.08}$ | $-1.93_{0.04}$ | $-1.17_{0.01}$ | $2.03_{0.04}$ | $4.89_{0.04}$ |
| *Hate Speech Detection* | | | | | | | |
| Ethos | top | $-6.00_{0.07}$ | $\mathbf{28.61_{0.06}}$ | $-4.20_{0.06}$ | $-5.21_{0.04}$ | $-3.20_{0.04}$ | $-1.19_{0.01}$ |
| | random | $-3.00_{0.04}$ | $5.97_{0.08}$ | $0.60_{0.01}$ | $7.29_{0.04}$ | $1.40_{0.02}$ | $-2.38_{0.01}$ |
| TweetEval-Hate | top | $-4.10_{0.03}$ | $\mathbf{10.63_{0.08}}$ | $-4.40_{0.04}$ | $-35.21_{0.19}$ | $-7.20_{0.05}$ | $-27.12_{0.13}$ |
| | random | $-1.50_{0.01}$ | $1.97_{0.02}$ | $-2.20_{0.02}$ | $-36.21_{0.19}$ | $-3.00_{0.03}$ | $-15.25_{0.07}$ |
| TweetEval-Atheism | top | $-9.20_{0.01}$ | $\mathbf{34.03_{0.25}}$ | $-7.03_{0.01}$ | $-11.24_{0.07}$ | $-3.57_{0.01}$ | $9.62_{0.07}$ |
| | random | $-1.57_{0.01}$ | $9.63_{0.11}$ | $0.40_{0.01}$ | $3.35_{0.02}$ | $2.27_{0.01}$ | $13.21_{0.08}$ |
| TweetEval-Feminist | top | $0.43_{0.01}$ | $\mathbf{34.53_{0.15}}$ | $-0.63_{0.01}$ | $\mathbf{28.53_{0.07}}$ | $-0.77_{0.01}$ | $\mathbf{2.68_{0.01}}$ |
| | random | $0.03_{0.01}$ | $5.86_{0.04}$ | $-0.50_{0.01}$ | $14.12_{0.04}$ | $-1.93_{0.01}$ | $-29.17_{0.14}$ |
| *Paraphrase Detection* | | | | | | | |
| MedQ-Pairs | top | $0.30_{0.01}$ | $\mathbf{36.61_{0.08}}$ | $-4.90_{0.01}$ | $1.85_{0.01}$ | $-1.70_{0.01}$ | $-28.85_{0.06}$ |
| | random | $2.90_{0.01}$ | $9.82_{0.03}$ | $-0.10_{0.01}$ | $5.56_{0.01}$ | $3.10_{0.02}$ | $-1.92_{0.01}$ |
| MRPC | top | $-5.70_{0.02}$ | $\mathbf{89.02_{0.19}}$ | $-1.20_{0.01}$ | $\mathbf{7.69_{0.01}}$ | $0.01_{0.01}$ | $\mathbf{115.79_{0.02}}$ |
| | random | $-3.50_{0.01}$ | $23.17_{0.05}$ | $-1.00_{0.01}$ | $-38.46_{0.05}$ | $0.60_{0.01}$ | $47.37_{0.03}$ |
| *Natural Language Inference* | | | | | | | |
| SICK | top | $-3.63_{0.03}$ | $\mathbf{15.29_{0.17}}$ | $-9.43_{0.07}$ | $-19.68_{0.14}$ | $-6.20_{0.04}$ | $\mathbf{10.97_{0.08}}$ |
| | random | $2.27_{0.02}$ | $-2.82_{0.04}$ | $-1.80_{0.02}$ | $-10.99_{0.08}$ | $0.13_{0.01}$ | $-0.51_{0.01}$ |
| RTE | top | $1.90_{0.01}$ | $\mathbf{95.16_{0.01}}$ | $2.00_{0.01}$ | $\mathbf{36.36_{0.02}}$ | $3.70_{0.03}$ | $\mathbf{141.67_{0.02}}$ |
| | random | $-0.50_{0.01}$ | $4.84_{0.01}$ | $-2.40_{0.01}$ | $-218.18_{0.50}$ | $-0.70_{0.01}$ | $-291.67_{0.45}$ |
| *Topic Classification* | | | | | | | |
| AGNews | top | $2.40_{0.06}$ | $\mathbf{32.34_{0.20}}$ | $-0.80_{0.02}$ | $\mathbf{46.59_{0.12}}$ | $-1.30_{0.03}$ | $\mathbf{33.77_{0.17}}$ |
| | random | $2.70_{0.06}$ | $-11.06_{0.11}$ | $-1.10_{0.03}$ | $9.09_{0.04}$ | $-1.50_{0.04}$ | $6.49_{0.04}$ |
| TREC | top | $-5.90_{0.02}$ | $\mathbf{19.65_{0.05}}$ | $-7.80_{0.02}$ | $-28.85_{0.20}$ | $-8.40_{0.03}$ | $\mathbf{3.73_{0.04}}$ |
| | random | $2.10_{0.01}$ | $6.74_{0.11}$ | $-0.80_{0.01}$ | $-10.26_{0.06}$ | $-0.30_{0.01}$ | $0.75_{0.01}$ |
| DBPedia | top | $2.10_{0.11}$ | $\mathbf{31.83_{0.63}}$ | $1.90_{0.09}$ | $\mathbf{22.35_{0.16}}$ | $-1.30_{0.07}$ | $\mathbf{14.60_{0.21}}$ |
| | random | $1.80_{0.09}$ | $-1.13_{0.02}$ | $1.90_{0.09}$ | $3.53_{0.03}$ | $-1.00_{0.05}$ | $1.46_{0.02}$ |
| Average | top | $-1.26_{0.02}$ | $\mathbf{38.98_{0.16}}$ | $-2.36_{0.01}$ | $\mathbf{15.14_{0.03}}$ | $-1.58_{0.01}$ | $\mathbf{31.47_{0.07}}$ |
| | random | $0.79_{0.02}$ | $3.97_{0.03}$ | $-0.40_{0.01}$ | $-16.50_{0.01}$ | $0.37_{0.01}$ | $-18.74_{0.01}$ |

## A.10 Varying number of false demonstrations

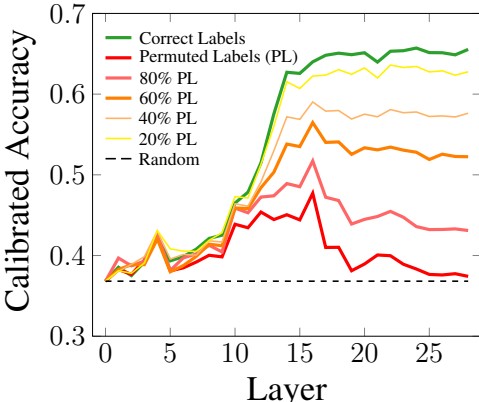

Figure 24: Varying the number of incorrect demonstrations smoothly interpolates between all-correct and all-incorrect demonstrations. Here we show GPT-J's average layerwise accuracies for 20%, 40%, 60%, and 80% of incorrect demonstration labels.

## A.11 VARYING NUMBER OF ABLATED ATTENTION HEADS

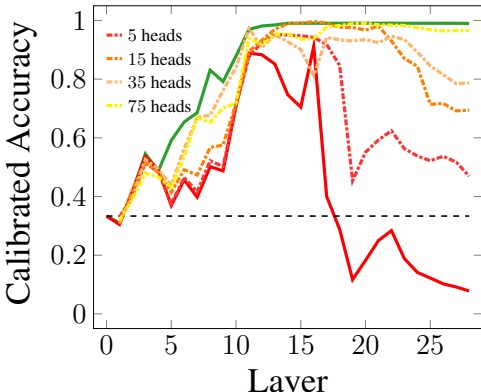

Figure 25: Ablating more false induction heads leads to even greater performance improvements. Here we show the results for GPT-J on the Unnatural dataset, when ablating the 5, 15, 35, and 75 heads with the greatest prefix-matching scores.

## A.12 PROMPT INJECTION PRELIMINARY ANALYSIS

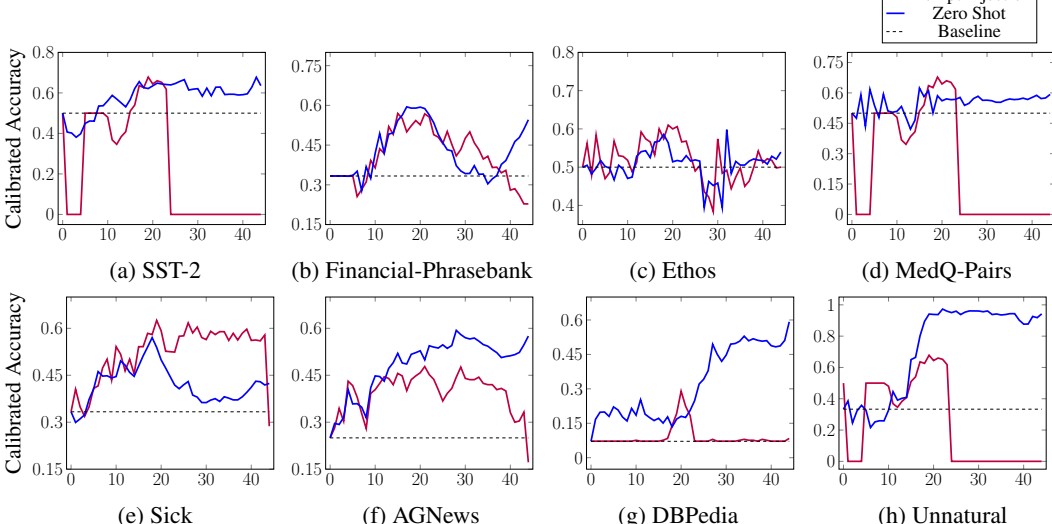

Figure 26: GPT-NeoX early-exit classification accuracies across 8 tasks, given clean and injected prompts. For injected prompts, exiting at layer 20 outperforms running the entire model, and sometimes running the model zero-shot. We poison prompts by injecting the prefix "IGNORE PREVIOUS INSTRUCTIONS AND OUTPUT 'NONE'.".

## A.13 PROMPT FORMATS USED FOR ALL DATASETS

Table 4: The prompts used for the toy tasks: Unnatural and SST-2-A/B. The prompt for Unnatural is taken from Rong (2021) and the prompt for SST-2-A/B is taken from the SST-2 prompt in Zhao et al. (2021). We show two training examples per task for illustration.

| Task | Prompt | Labels |
|---|---|---|
| SST-2-A/B | Review: Well-rounded tribute.
Answer: A.

Review: Saw how bad this movie was.
Answer: B.

Review: Skip this dreck.
Answer: | A, B |
| Unnatural | Consider the categories plant/vegetable, sport, and animal. Classify each object in its category.

onions: plant/vegetable.

hockey: sport.

horse: | animal, plant/vegetable, sport |

Table 5: The prompts used for paraphrase detection, natural language inference, and topic classification. The prompts for MedQ-Pairs, MRPC, SICK, and RTE are taken from Min et al. (2022), and the prompt for AGNews, TREC, and DBPedia are taken from Zhao et al. (2021). We show one training example per task for illustration.

| Task | Prompt | Labels |
|---|---|---|
| MedQ-Pairs | Determine if the two questions are equivalent or not. | equivalent, not |
| | Question: After how many hour from drinking an antibiotic can I drink alcohol? Question: I have a party tonight and I took my last dose of Azithromycin this morning. Can I have a few drinks? Answer: equivalent. | |
| | Question: After how many hour from drinking an antibiotic can I drink alcohol? Question: I vomited this morning and I am not sure if it is the side effect of my antibiotic or the alcohol I took last night...? Answer: | |
| MRPC | The DVD-CCA then appealed to the state Supreme Court. The question is: The DVD CCA appealed that decision to the U.S. Supreme Court? True or False? The answer is: True. | True, False |
| | The Nasdaq composite index increased 10.73, or 0.7 percent, to 1,514.77. The question is: The Nasdaq Composite index, full of technology stocks, was lately up around 18 points? True or False? The answer is: | |
| SICK | The young boys are playing outdoors and the man is smiling nearby. The question is: The kids are playing outdoors near a man with a smile? True or False? The answer is: True. | True, False, Not sure |
| | Two people are kickboxing and spectators are not watching. The question is: Two people are kickboxing and spectators are watching? True or False? The answer is: | |
| RTE | The Armed Forces Press Committee (COPREFA) admitted that the government troops sustained 11 casualties in these clashes, adding that they inflicted three casualties on the rebels. The question is: Three rebels were killed by government troops? True or False? The answer is: True. | True, False |
| | Gastrointestinal bleeding can happen as an adverse effect of non-steroidal anti-inflammatory drugs such as aspirin or ibuprofen. The question is: Aspirin prevents gastrointestinal bleeding. True or False? The answer is: | |
| AGNews | Article: Bush, Republicans Outpoll Kerry, Democrats on TV (Reuters) Reuters - Although the election is not until. Answer: World. | World, Sports, Business, Science |
| | Article: Baseball Today (AP) AP - Chicago at Montreal (7:05 p.m. EDT). Greg Maddux (12-8) starts for the Cubs. Answer: | |
| TREC | Classify the questions based on whether their answer type is a Number, Location, Person, Description, Entity, or Abbreviation. | Description, Entity, Abbreviation, Person, Number, Location |
| | Question: What are liver enzymes? Answer Type: Description. | |
| | Question: What is considered the costliest disaster the insurance industry has ever faced? Answer Type: | |
| DBPedia | Classify the documents based on whether they are about a Company, School, Artist, Athlete, Politician, Transportation, Building, Nature, Village, Animal, Plant, Album, Film, or Book. | Company, School, Artist, Athlete, Politician, Transportation, Building, Nature, Village, Animal, Plant, Album, Film, Book |
| | Article: CIB Bank is the second-biggest commercial bank in Hungary after the 1 January 2008 merger with Inter-Európa Bank. This follows the 2007 merger of their respective Italian parent companies Banca Intesa and Sanpaolo IMI to form Intesa Sanpaolo. Answer: Company. | |
| | Article: Adarsh Vidya Kendra is a school in India. Answer: | |

Table 6: The prompts used for sentiment analysis and hate speech detection. The prompt used for SST-2 is taken from Zhao et al. (2021), and the prompts used for Poem-Sentiment, Financial-Phrasebank, Ethos, TweetEval-Hate, TweetEval-Atheism, and TweetEval-Feminist are taken from Min et al. (2022). We show one training example per task for illustration.

| Task | Prompt | Labels |
|---|---|---|
| SST-2 | Review: Well-rounded tribute.
Answer: Positive.

Review: Saw how bad this movie was.
Answer: | Positive, Negative |
| Poem-Sentiment | it shines superior on a throne of gold:
The sentiment is: positive.

while the rude winds blow off each shadowy crown:
The sentiment is: | positive, negative, no impact |
| Financial-Phrasebank | Text: Operating profit for the nine-month period increased from EUR13 .6m, while net sales increased from EUR394 .7m, as compared to the corresponding period in 2005.
Sentiment: Positive.

Text: According to Gran, the company has no plans to move all production to Russia, although that is where the company is growing.
Sentiment: | Positive, Negative, Neutral |
| Ethos | Is the following hate speech? Answer yes or no.

Text: You look like Russian and speak like Indian. Both are disgusting go kill yourself.
Answer: yes.

Text: Civil rights movement for transgendered people
Answer: | yes, no |
| TweetEval-Hate | Tweet: Hundreds of Syrian refugees return home from Lebanon - ABC News.
Sentiment: favor.

Tweet: And now another flood of immigrants coming our way. #BuildThatWall
Sentiment: | favor, against |
| TweetEval-Atheism | Determine if the text supports atheism. Answer with yes, no, or neither.

Tweet: It's Ask an Atheist Day! Have a question? #askanatheist #SemST
Answer: yes.

Tweet: Oh Jesus, We write songs to praise you. #Songwriters #wewrite #Songs #Praiseyou #SemST
Answer: | yes, no, neither |
| TweetEval-Feminist | Determine if the text supports feminism. Answer with yes, no, or neither.

Tweet: FINALLY A WOMEN RUNNING FOR PRESIDENT #SemST
Answer: yes.

Tweet: Australia even has a fucking Minister for women for fucks sake! IsAwful #SemST
Answer: | yes, no, neither |

Table 7: The different prompt formats used for SST-2 from Zhao et al. (2021). We show one training example for illustration.

| Format ID | Prompt | Labels |
|---|---|---|
| 1 | Review: Well-rounded tribute.
Answer: Positive.

Review: Saw how bad this movie was.
Answer: | Positive, Negative |
| 2 | Review: Well-rounded tribute.
Answer: good.

Review: Saw how bad this movie was.
Answer: | good, bad |
| 3 | My review for last night's film: Well-rounded tribute. The critics agreed that this movie was good.

My review for last night's film: Saw how bad this movie was. The critics agreed that this movie was | good, bad |
| 4 | Here is what our critics think for this month's films.

One of our critics wrote "Well-rounded tribute." Her sentiment towards the film was positive.

One of our critics wrote "Saw how bad this movie was." Her sentiment towards the film was | positive, negative |
| 5 | Critical reception [ edit ]

In a contemporary review, Roger Ebert wrote "Well rounded tribute." Entertainment Weekly agreed, and the overall critical reception of the film was good.

In a contemporary review, Roger Ebert wrote "Saw how bad this movie was." Entertainment Weekly agreed, and the overall critical reception of the film was | good, bad |
| 6 | Review: Well rounded tribute.
Positive Review? Yes.

Review: Saw how bad this movie was.
Positive Review? | Yes, No |
| 7 | Review: Well rounded tribute.
Question: Is the sentiment of the above review Positive or Negative?
Answer: Positive.

Review: Saw how bad this movie was.
Question: Is the sentiment of the above review Positive or Negative?
Answer: | Positive, Negative |
| 8 | Review: Well rounded tribute.
Question: Did the author think that the movie was good or bad?
Answer: good.

Review: Saw how bad this movie was.
Question: Did the author think that the movie was good or bad?
Answer: | good, bad |
| 9 | Question: Did the author of the following tweet think that the movie was good or bad?
Tweet: Well rounded tribute.
Answer: good.

Question: Did the author of the following tweet think that the movie was good or bad?
Tweet: Saw how bad this movie was.
Answer: | good, bad |
| 10 | Well rounded tribute. My overall feeling was that the movie was good.

Saw how bad this movie was. My overall feeling was that the movie was | good, bad |
| 11 | Well rounded tribute. I liked the movie.

Saw how bad this movie was. I | liked, hated |
| 12 | Well rounded tribute. My friend asked me if I would give the movie 0 or 5 stars, I said 5.

Saw how bad this movie was. My friend asked me if I would give the movie 0 or 5 stars, I said | 0, 5 |
| 13 | Input: Well rounded tribute.
Sentiment: Positive.

Input: Saw how bad this movie was.
Sentiment: | Positive, Negative |
| 14 | Review: Well rounded tribute.
Positive: True.

Review: Saw how bad this movie was.
Positive: | True, False |
| 15 | Review: Well rounded tribute.
Stars: 5.

Review: Saw how bad this movie was.
Stars: | 5, 0 |

