# OpenReview forum: "Overthinking the Truth: Understanding how Language Models Process False Demonstrations"
_ICLR.cc/2024/Conference — ICLR 2024 spotlight_

### Official Review · Reviewer_Lx2y · 2023-10-21

**Soundness:** 3 good
**Presentation:** 3 good
**Contribution:** 3 good
**Rating:** 6
**Confidence:** 5

**Summary:**

This paper studies inner mechanism of LLMs when facing false demonstations.

**Strengths:**

I like this paper. This work provides interesting findings.

**Weaknesses:**

1. Since the authors conduct experiments under in-context learning setting, we may not be sure these findings are from models themself or implicit tuning (in-context learning). Could the authors provide explanations?

2. Can we expand tasks to include generation-based tasks like QA?

**Questions:**

Please see Weaknesses.

---

> ### Author Response · Authors · 2023-11-21
> **Response #3**
>
> Thank you for your valuable feedback. We hope to address your concerns and answer your questions.
>
> > “Since the authors conduct experiments under in-context learning setting, we may not be sure these findings are from models themself or implicit tuning (in-context learning). Could the authors provide explanations?”
>
> Our understanding is that you are asking how our results would transfer to settings other than in-context learning.
> While most of our experiments are on ICL, we also found evidence of overthinking given misleading prompts in non-ICL settings, for example for prompt injection (Figure 26). Therefore we believe that the phenomena we uncovered generalize beyond ICL.
>
> > “Can we expand tasks to include generation-based tasks like QA?”
>
> In our paper we focused on text classification tasks, since they allowed us to run more controlled experiments. However, our results can in principle be extended to generation tasks. For example, it would be valuable to apply our methods to study code completion models prompted with bug-ridden code [1], or prompts containing factual inaccuracies [2]. Early decoding at the critical layer could improve performance in these cases as well.
>
> [1] Perry et al. 2022, Do Users Write More Insecure Code with AI Assistants?
>
> [2] Burns et al. 2022, Discovering Latent Knowledge in Language Models Without Supervision

---

### Official Review · Reviewer_bUf7 · 2023-10-30

**Soundness:** 3 good
**Presentation:** 3 good
**Contribution:** 4 excellent
**Rating:** 8
**Confidence:** 3

**Summary:**

The paper focuses on understanding how language models process correct versus incorrect inputs in few-shot learning contexts, emphasizing harmful imitation. Two phenomena are identified: overthinking and false induction heads. Overthinking occurs when models, given incorrect demonstrations, tend to have reduced accuracy when predictions are decoded from later layers; stopping the model early, before all layers have processed the input, can increase accuracy. False induction heads are specific components in the model that contribute to overthinking by attending to and replicating incorrect information from previous demonstrations. The study's findings advocate for a deeper examination of intermediate model computations to understand and mitigate harmful behaviors in language models​.

**Strengths:**

- The paper explores new concepts like "overthinking" (previously introduced) and "false induction heads" (novel), offering a fresh lens to study transformers, particularly in handling harmful imitation.

- Solid research execution is evident. Specific model components, like attention heads causing errors, are identified and well analyzed.

- The paper is well-structured and very readable. Concepts are clearly and consistently articulated.

- The methodologies provided comprise a strong set of interpretability methods directed specifically at in-context learning and the role of the examples included.
  - For instance, interpreting outputs from intermediate layers in such ways may prove useful for understanding what the optimal sets of example to provide to the LLM is, in addition to what is attempted in this paper, which is to understand how bad examples affect overall accuracy.

- The paper studies the important issue of harmful content (which, as outlined above, may extend eventually to non-useful or even non-optimal content within the context). Overall, the findings are practically significant and offer insights for controlling model outputs against harmful examples and wrongful imitation.

**Weaknesses:**

- The paper details the phenomena of overthinking and false induction heads, but it doesn’t fully articulate how these insights could be applied practically to improve model behavior.
  - For instance, while head ablations are discussed as a method to reduce overthinking, the practical impact of these modifications on model functionality and broader applicability is not thoroughly explored in real-world scenarios​ i.e. how does one know when the input examples are likely to be false, and that heads ought to be ablated?
- The methodology involves decoding from intermediate layers and identifying false induction heads, but the paper could provide a more explicit explanation of these processes i.e. how predictions were extracted and analyzed at each layer, and how the heads contributing to overthinking were identified and analyzed.
- The paper could better acknowledge that the 'incorrectness' of permuted labels can vary based on the dataset and what each label signifies. Some permuted labels might be clearly wrong, while others could be more ambiguous, and providing specific examples for each dataset could clarify this aspect and help to understand the results.
  - It's not clear in the example from Figure 1 that demonstrations are "incorrect", and that all "imitation" is wrongful- there is still an argument that context is important, and in this example (at least in the example where all labels are permuted), could the LLM not understand the label permutation and simply interpret that negative sentiments require a positive classification, and vice versa?
- The reference and justification around the use of the logit lens near the start of the paper could be strengthened, though the authors do come back to discuss it in more detail towards the end.

**Questions:**

Please see weaknesses.

---

> ### Author Response · Authors · 2023-11-21
> **Response #2**
>
> Thank you for your helpful review.
>
> > "The paper details the phenomena of overthinking and false induction heads, but it doesn’t fully articulate how these insights could be applied practically to improve model behavior. For instance, while head ablations are discussed as a method to reduce overthinking, the practical impact of these modifications on model functionality and broader applicability is not thoroughly explored in real-world scenarios​ i.e. how does one know when the input examples are likely to be false, and that heads ought to be ablated?"
>
> While we were mainly focused on understanding how models process inaccurate prompts, we believe our results suggest some directions for mitigation. One way to identify if input examples are false might be to look for sharp changes in prediction during the model’s processing. Another possibility would be to train a probe on the model activations to determine if the input examples are false. Moreover, in our experiments we found that ablating the heads / decoding early led to large improvements given incorrect demonstrations, with no or much smaller effects given correct demonstrations. Therefore, at least in our ICL setting, early exiting improved performance without needing to determine which input examples are false.
>
> > “The methodology involves decoding from intermediate layers and identifying false induction heads, but the paper could provide a more explicit explanation of these processes i.e. how predictions were extracted and analyzed at each layer, and how the heads contributing to overthinking were identified and analyzed.”
>
> Thank you for this feedback. Here is a more detailed explanation, which we can add to the paper if you find it clear:
>
> To extract predictions from each layer, we used the logit lens (Nostalgebraist, 2020): we got logits by applying the unembedding matrix to the post layernorm activations after each layer (equation on page 5).
>
> We chose the 5 heads with the highest context following scores (equation on page 7). We selected the heads on our “Unnatural” synthetic dataset to demonstrate that these heads generalize to more realistic datasets.
>
> Please let us know if there is anything further we can clarify in this procedure.
>
> > “The paper could better acknowledge that the 'incorrectness' of permuted labels can vary based on the dataset and what each label signifies. Some permuted labels might be clearly wrong, while others could be more ambiguous, and providing specific examples for each dataset could clarify this aspect and help to understand the results.”
>
> Our understanding of what you mean by ambiguous is that in some datasets, some labels have similar meanings. For example, in DBPedia, the class labels “Plant” and “Nature” are similar, so assigning the label “Nature” to a plant would not be clearly wrong.
>
> Assuming this is what you had in mind, this ambiguity generally should not affect our conclusions, since we use a new permutation for each input sequence and so most permutations send “Plant” to a different, unrelated label. Therefore, this effect gets averaged out.

---

### Official Review · Reviewer_qL6Z · 2023-10-31

**Soundness:** 3 good
**Presentation:** 3 good
**Contribution:** 3 good
**Rating:** 8
**Confidence:** 3

**Summary:**

The paper investigates how language models can learn to generate harmful outputs from few-shot prompts containing inaccuracies or biases.

The key findings are:

Models tend to "overthink" and decrease in accuracy on incorrect prompts after a certain "critical layer", while accuracy keeps improving on correct prompts. This suggests incorrect information mainly affects later processing stages.
Attention heads in later layers preferentially attend to and reproduce incorrect labels from the prompt demonstrations. Ablating a small number of these "false induction heads" significantly reduces the accuracy gap between correct and incorrect prompts.

**Strengths:**

The paper:
- Provides novel insights into the internals of in-context learning through layerwise decoding and attention analysis. Links model behavior to specific components.

- Extensive experiments across models, datasets and prompt variations. Head ablation results generalize across tasks.

- Connects to related ideas like overthinking and induction heads. Builds understanding of model internals.

**Weaknesses:**

The paper lacks in the following ways:
- Focuses on a simplified setting of permutations of class labels.
- Does not cover more subtle inaccuracies or biases.
- Only studies text classification tasks. Unclear if findings apply to more open-ended generative tasks.
- No modification of model training process itself to mitigate issues.

**Questions:**

- If the prompt contained factual inaccuracies or harmful content, would you expect to see similar overthinking and false induction effects?
- Could you train with additional regularization to discourage attention to incorrect prompt content?

---

> ### Author Response · Authors · 2023-11-20
> **Response #1**
>
> Thank you for your helpful review.
>
> > “Focuses on a simplified setting of permutations of class labels.”
> > “If the prompt contained factual inaccuracies or harmful content, would you expect to see similar overthinking and false induction effects?”
>
> In addition to demonstrations containing permuted class labels, we also obtained results for demonstrations with random labels as well as prompts in which only half of demonstrations have permuted labels (Figure 4).
>
> Furthermore, we show some preliminary results beyond the in-context learning setting: Figure 26 shows that overthinking can also occur in prompt injection attacks, and that early exiting can mitigate the attack (Figure 26).
>
> Therefore, we also expect overthinking and false induction to occur for other prompts containing factual inaccuracies or harmful content. For example, we may find overthinking in code completion models given bug-ridden prompts.
>
> > “Could you train with additional regularization to discourage attention to incorrect prompt content?”
>
> We found that ablating heads that attend to incorrect prompt tokens improved performance. This suggests that adding an auxiliary loss specifically penalizing high attention scores to tokens in incorrect demonstrations might help mitigate false context following.

---

### Meta-Review · Area_Chair_Epaz · 2023-12-05

**Metareview:**

This paper provides new and interesting insights about the mechanisms by which incorrect demonstrations affect model prediction. While the “overthinking” term has been coined before to refer to “when a DNN can reach correct predictions before its final layer.“ and has been reported in CNNs [1], the authors do a great job in studying it further in the context of incorrect demonstrations in Transformer language models. In particular, they experiment with two in-context learning setups: random answers, and permutation of half the demonstrations. Their findings suggest there are specific layers after which the accuracy keeps decreasing. They try to explain this phenomenon further by looking into attention heads, and identifying heads in the later layers that copy incorrect information, “false induction heads”, and showing that ablating them can undo the damage done by the wrong demonstrations. While the paper mainly focuses on in-context learning setups, they also provide preliminary results in the appendix on a prompt injection attack setup.

The experiments are scientifically sound and the paper is clear, well-written, and easy to follow.

[1] Kaya, Y., Hong, S., & Dumitras, T. (2019, May). Shallow-deep networks: Understanding and mitigating network overthinking. In International conference on machine learning (pp. 3301-3310). PMLR.

**Justification For Why Not Higher Score:**

This paper provides interesting insights about how incorrect demonstrations influence model predictions, is well-written, and the experiments are well-designed and thorough. The fact that information gets processed gradually across layers has been reported in other setups, and this can be seen as a special case of such reported findings. Also the role of induction heads is well-known. Arguably, emergence of "false induction heads" is a natural implication of the experimental setups studied in this paper. Therefore, the scale of the contributions of this paper may not be significant enough to warrant an oral presentation. I still find this work interesting and worthwhile. So if the SAC/PCs would like to offer an oral presentation to it, I would not be against that.

**Justification For Why Not Lower Score:**

This paper provides interesting insights, especially for the mechanistic interpretability community, and can have implications for applications on controllability further down the road. Therefore, it deserves more than a regular poster presentation.

---

### Decision · Program_Chairs · 2024-01-16

Accept (spotlight)